# Evolution of SARS-CoV-2 in white-tailed deer in Pennsylvania 2021–2024

Andrew D. Marques[ID][1], Matthew Hogenauer[1], Natalie Bauer[2], Michelle Gibison[2], Beatrice DeMarco[2], Scott Sherrill-Mix[3], Carter Merenstein[1], Ronald G. Collman[4], Roderick B. Gagne[2]*, Frederic D. Bushman[ID][1]*

1 Department of Microbiology, Perelman School of Medicine, University of Pennsylvania, Philadelphia, Pennsylvania, United States of America, 2 Department of Pathobiology, Wildlife Futures Program, University of Pennsylvania School of Veterinary Medicine, New Bolton Center, Kennett Square, Pennsylvania, United States of America, 3 Department of Microbiology, Genetics, and Immunology, College of Veterinary Medicine, Michigan State University, East Lansing, Michigan, United States of America, 4 Division of Pulmonary, Allergy, and Critical Care, Philadelphia, Pennsylvania, United States of America

* bushman@pennmedicine.upenn.edu (FDB); rgagne@vet.upenn.edu (RBG)

## Abstract

SARS-CoV-2 continues to transmit and evolve in humans and animals. White-tailed deer (*Odocoileus virginianus*) have been previously identified as a zoonotic reservoir for SARS-CoV-2 with high rates of infection and probable spillback into humans. Here we report sampling 1,127 white-tailed deer (WTD) in Pennsylvania, and a genomic analysis of viral dynamics spanning 1,017 days between April 2021 and January 2024. To assess viral load and genotypes, RNA was isolated from retropharyngeal lymph nodes and analyzed using RT-qPCR and viral whole genome sequencing. Samples showed a 14.64% positivity rate by RT-qPCR. Analysis showed no association of SARS-CoV-2 prevalence with age, sex, or diagnosis with Chronic Wasting Disease. From the 165 SARS-CoV-2 positive WTD, we recovered 25 whole genome sequences and an additional 17 spike-targeted amplicon sequences. The viral variants identified included 17 Alpha, 11 Delta, and 14 Omicron. Alpha largely stopped circulating in humans around September 2021, but persisted in WTD as recently as March of 2023. Phylodynamic analysis of pooled genomic data from Pennsylvania documents at least 12 SARS-CoV-2 spillovers from humans into WTD, including a recent series of Omicron spillovers. Prevalence was higher in WTD in regions with crop coverage rather than forest, suggesting an association with proximity to humans. Analysis of seasonality showed increased prevalence in winter and spring. Multiple examples of recurrent mutations were identified associated with transmissions, suggesting WTD-specific evolutionary pressures. These data document ongoing infections in white-tailed deer, probable onward transmission in deer, and a remarkable rate of new spillovers from humans.

## Author summary

SARS-CoV-2 continues to evolve and circulate in humans and animals. Sometimes accumulating genetic changes give rise to new viral variants, the origins of which are

**Data availability statement:** Viral sequences published in this study can be accessed through GISAID and NCBI using the accessions listed in S2 Table. Raw data can be accessed through SRA accession (https://www.ncbi.nlm.nih.gov/sra?LinkName=bioproject_sra_all&from_uid=1134208) or with BioProject accession PRJNA1134208. Previously published consensus sequences used in this work have accessions that can be found in S3 Table. Code can be accessed on GitHub at https://github.com/andrewdmarques/PA_WTD_Genomics.

**Funding:** RGC and FDB are supported by institutional core funding by the Penn University Research Foundation (https://research.upenn.edu/funding-opportunity/university-research-foundation-research-grant-2/), the Penn Center for Global Genomics and Health Equity Keystone Pilot Grant (grant ID GGHE-KP-2021-001, https://globalgenomics.med.upenn.edu/updatedWebsite2021_gghe/keystone.php), and the Penn Center for Research on Coronaviruses and Other Emerging Pathogens (https://www.pennmedicine.org/research-at-penn/research-specialty-areas/penn-research-programs-and-interests/coronavirus-research-center). Further funding for RGC and FDB come from the National Institutes of Health (grant ID R61/33-HL137063, https://www.nih.gov/). RGC is supported by the Penn Center for AIDS Research (grant ID P30-AI045008 https://www.med.upenn.edu/cfar/). The funders had no role in study design, data collection and analysis, decision to publish, or preparation of the manuscript.

**Competing interests:** The authors have declared that no competing interests exist.

incompletely understood and may involve transmissions between humans and animals. White-tailed deer (*Odocoileus virginianus*) is one of several host species known to harbor SARS-CoV-2 at high prevalence. We thus studied circulation and evolutionary dynamics of SARS-CoV-2 in 1,127 white-tailed deer in Pennsylvania from 2021 to 2024. Prevalence was higher in white-tailed deer in regions with crop coverage rather than forest. Analysis of seasonality showed increased prevalence in winter and spring. Our viral genome sequence analysis identified white-tailed deer as a SARS-CoV-2 reservoir of ancestral and highly divergent lineages consistent with genetic distances similar to new variants that have emerged in the human population. We found multiple examples of recurrent mutations in independent suspected spillover events, consistent with selective pressures in white-tailed deer. Together, these findings provide a unique longitudinal perspective on SARS-CoV-2 circulation, persistence, and evolution in Pennsylvania wild white-tailed deer.

## Introduction

COVID-19 is caused by severe acute respiratory syndrome coronavirus 2 (SARS-CoV-2). SARS-CoV-2 is highly transmissible among humans and has been responsible for more than 750 million reported cases and 7 million deaths globally since its emergence in late 2019 [1]. In addition to humans, both wild and domestic animals have been reported to be susceptible, including infections documented in more than 55 vertebrate species [2–24] (S1 Table). There are credible reports of spill-back into humans for four of these species, including from wild white-tailed deer (WTD), domestic hamster, domestic cat, and farmed mink [11,25–27]. These findings focus intense interest on understanding potential variants that may emerge from SARS-CoV-2 infection in non-human vertebrates, and the consequences for spill-back into humans.

SARS-CoV-2 evolution often involves genetic jumps yielding variant viruses with more genetic changes per unit time than expected from conventional human-to-human transmission. The origin of these new variants is not well understood. One origin may be prolonged infection of immunocompromised patients; another possibility is viral divergence in an animal species followed by spillback into humans [25,28–31]. Data in favor of the first theory include the documented emergence of substitutions conferring drug-resistance, increased viral replication and immune evasion in immunocompromised patients with prolonged infections [30,32–36]. Several groups have reported measuring increased rates of within-host evolution in immunocompromised subjects [29,32,33,37–39]. However, coronaviruses are known to evolve in animal reservoirs, as indicated by studies of the emergence of SARS in 2003 and MERS in 2012 from bats and camels [40,41].

Among possible sources of spillback from animals, WTD remain one of the most important species to examine because of the high infection rates and abundant populations in close proximity to humans. It is estimated that there are more than one million WTD in Pennsylvania alone [42,43]. Since 2020, positivity rates have been reported as high as 35.8% in some parts of the United States [44]. We and others have previously reported that specific SARS-CoV-2 variants persisted in WTD populations for months after circulation in humans had ceased [45,46]. Several studies suggest that lineages specific to WTD subsequently appeared in isolated human cases, suggesting spill back from WTD [25,46]. Thus it is critical to monitor the large WTD viral reservoir for evolution of new variants and possible onward transmission of evolved variants to humans.

WTD do not display overt signs of infection when infected by SARS-CoV-2 [47,48]. Experimental inoculation of fawns resulted in no obvious signs of infection although a transient increase in body temperature was observed in most deer lasting through the first day [47]. Infectious viral particles were detected between days 2 and 5 after infection and viral RNA was detected between days 2 and 22 after infection. Many tissues including retropharyngeal lymph nodes were found to be RT-qPCR positive for viral RNA during infection [47].

Chronic wasting disease (CWD) is a prion-associated disease that may be a risk factor for SARS-CoV-2 in deer. CWD is a transmissible spongiform encephalopathy that affects the nervous system, takes about 16 months from exposure to symptom onset, and is invariably fatal to deer. CWD-positive deer show progressive weight loss, excessive urination, excessive salivation, and behavioral changes including stumbling and loss of fear of humans. To date, little is known about the possible risk factors associated with CWD and SARS-CoV-2, and whether the prion disease could contribute to the rise of new SARS-CoV-2 variants. It is uninvestigated whether CWD results in higher SARS-CoV-2 positivity rates as a result of increased human-deer interactions caused by a loss of fear of humans. It is also not known whether the malnourished CWD state could result in an altered immune response which may affect viral replication and evolution.

Here, we examine SARS-CoV-2 in Pennsylvania from 2021 to 2024 over 1,127 newly collected WTD specimens, including an investigation of the possible interplay between CWD and SARS-CoV-2.

## Results

### SARS-CoV-2 prevalence in WTD in Pennsylvania

We sampled 1,127 WTD retropharyngeal lymph nodes (RPLNs) collected from deer between 4/5/2021 and 1/17/2024 from 39 counties in Pennsylvania (S2 Table). These data complement and extend our previous study (n = 123 WTD samples) [45]. RPLNs are one of many sites known to harbor SARS-CoV-2 RNA in infected animals [47], and are a sample of convenience because RPLNs are among the tissues collected in the course of monitoring for CWD [47]. Cause of death for sampled animals included hunting (n = 414), roadkill (n = 731), and targeted removal (n = 12). Positivity was assessed through reverse-transcription quantitative polymerase chain reaction (RT-qPCR) using two different primers targeting distinct regions within the SARS-CoV-2 nucleocapsid [49]. We identified a total of 165/1,127 RPLN samples that were SARS-CoV-2 positive by RT-qPCR, for a total of 14.64% positive (95% confidence interval (CI) 12.7–16.9%).

Age was not significantly associated with SARS-CoV-2 positivity when comparing across groups of fawns (age 0-12 months, n = 128), yearlings (age 13-24 months, n = 549), and adults (age 25 months or older, n = 396) (Pearson's chi-squared p = 0.92) (Fig 1A). Looking only at positive samples, age was also not significantly associated with differences in Ct values (ANOVA p = 0.41) (Fig 1B).

Cause of death was significantly associated with SARS-CoV-2 positivity when comparing hunter harvested (n = 340), roadkill (n = 710), and targeted removal (n = 12) (Pearson's chi-squared p < 0.0001) (Fig 1C). A post-hoc pairwise comparison of proportions adjusting for multiple comparisons using Bonferroni correction suggested that targeted removals had a significantly higher positivity rate than hunter harvested (p < 0.0001), and roadkill (p < 0.0001) WTD. Targeted removals represent only a small subset (n = 12) of all WTD collected (n = 1,127). Eleven WTD were removed from the same county on the same day, and 10 of these tested positive for SARS-CoV-2, indicating removal of an infected cluster of animals which drove the statistical significance. Causes of death showed no association with SARS-CoV-2 Ct values (ANOVA p = 0.57) (Fig 1D).

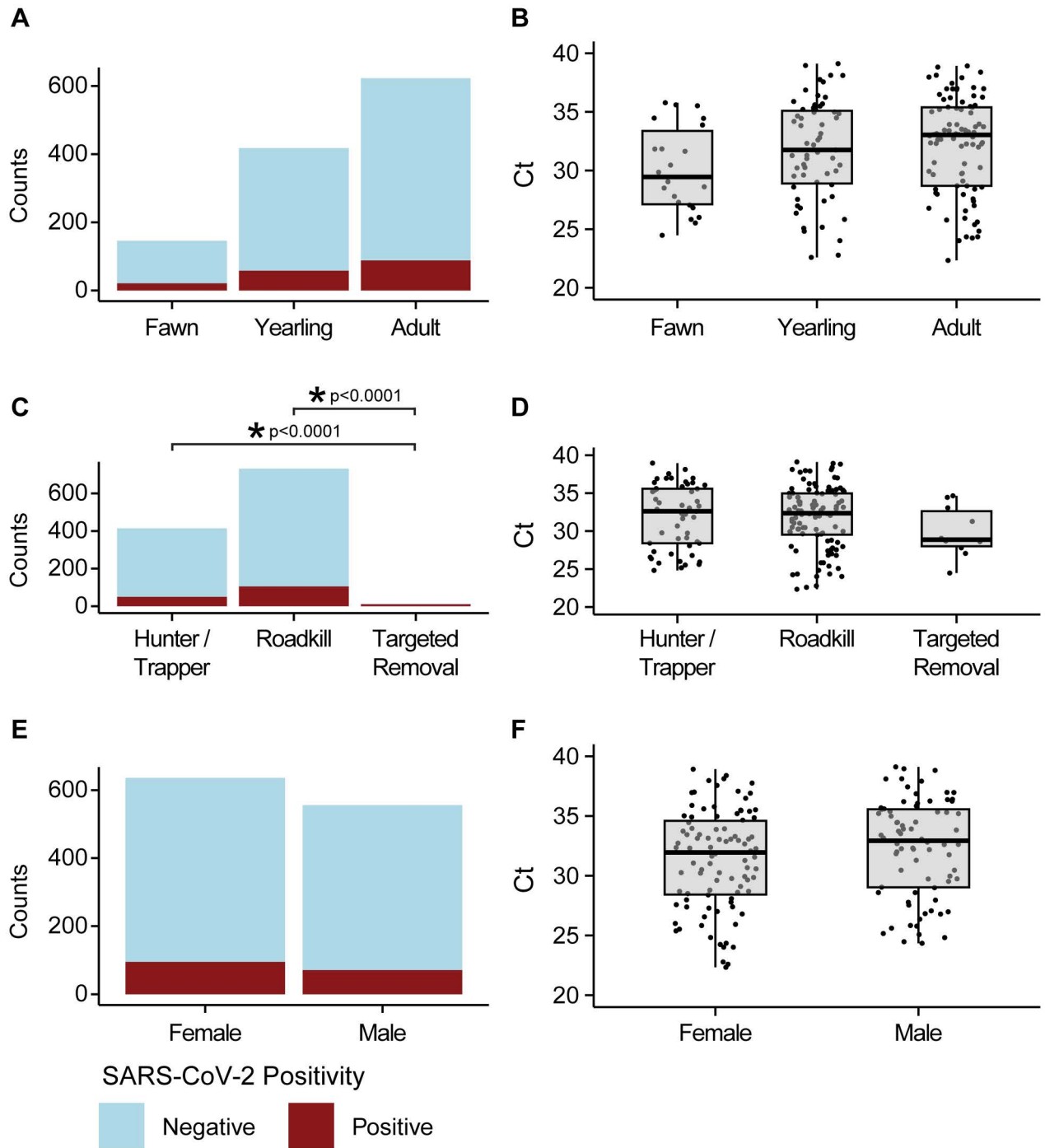

**Fig 1. RT-qPCR results for pooled Pennsylvania WTD samples from 2021–2024.** (A) Counts of WTD stratified by age. Categories include fawn, yearling, and adult, and are colored by RT-qPCR results. Here and below, blue indicates RT-qPCR negative and red indicates RT-qPCR positive animals. (B) Ct values of positive samples displayed with box plots for fawn, yearling, and adult WTD. (C) RT-qPCR counts stratified by cause of death (hunter harvested or trapped, roadkill, or targeted removal). (D) Ct values of the RT-qPCR positive WTD stratified by cause of death. (E) RT-qPCR counts stratified by female or male. (F) Ct values for RT-qPCR positive WTD.

The animal's sex was not significantly associated with SARS-CoV-2 positivity (males: n = 490; females: n = 587; 2-sample test for equality of proportions with continuity correction p = 0.4249, males = 65/490 and females = 89/587) (Fig 1E). Animal sex was also not significantly associated with average SARS-CoV-2 Ct (Welch 2-sample t-test p = 0.1132) (Fig 1F). The mean Ct for males was 32 (standard deviation (sd) = 3.94) and for females was 31 (sd = 3.80).

CWD samples were collected primarily during the hunting season over the course of 3 years (2021: September-December; 2022: October-December; and 2023: August-November). When comparing CWD-negative samples to CWD-positive samples collected at the same times, there was no significant difference in SARS-CoV-2 positivity (Two-sample test for equality of proportions with continuity correction p = 0.38) (Table 1 and S1 Fig). Among SARS-CoV-2 positive samples, there was no significant relationship between CWD status and Ct value (Welch two-sample t-test p = 0.18).

## Seasonality of Pennsylvania WTD SARS-CoV-2 positivity

Aggregating our RT-qPCR data from 2021 to 2024 (Fig 2), including samples from our previous publication [45] and those newly determined here, showed significant differences in SARS-CoV-2 positivity by season (Pearson's Chi-squared p < 0.0001) comparing spring (n = 124, March 19-June 20), summer (n = 41, June 21-September 21), fall (n = 706, September 22-December 20), and winter (n = 209, December 21-March 18) (Fig 2A). We also observed a significant difference in average Ct value by season (ANOVA p = 0.0323), where samples taken in winter had significantly higher viral RNA levels (lower Ct) than positive SARS-CoV-2 samples in the fall (Tukey multiple comparisons of means p = 0.016) (Fig 2B).

To probe this further, we developed a Bayesian logistic regression model to determine the estimated effect sizes of multiple variables, including fixed variables (season, ambient temperature during collection, and sex), and a random effect for cause of death. We include animal sex in our model to examine the model's performance with a variable that we would not expect to have an effect. As expected, sex did not correlate with SARS-CoV-2 positivity (males showed a 0.96 difference in odds more likely to be positive than females, 95% credible interval (CrI) (0.66-1.36)). Average weekly temperature from the time of sample collection (for the region in which the sample was collected) was used to determine whether ambient temperature might contribute to positivity rate, possibly by modulating sample degradation (e.g. samples collected during warmer weeks might have lower positivity rate due to increased RNA degradation) (S2 Fig). Temperature had no predictive effect on positivity (odds ratio = 0.97, 95% CrI 0.93–1.01). Season had a significant correlation with SARS-CoV-2 positivity with winter having 2.25-fold increased odds (95% CrI 1.47–3.47) compared to fall, and spring having a 2.06-fold increased odds (95% CrI 1.21–3.50) compared to fall. Summer and fall did not show a credible difference for positivity rates from each other.

To examine effects of regional ambient temperature for sample collection sites further, we ran a secondary analysis comparing samples collected at temperatures above and below 10°C. 11.6% of samples collected in regions with ambient temperatures above 10°C were SARS-CoV-2 positive (n = 232 WTD), compared to 14.9% of samples collected in regions

**Table 1. Pooled CWD-positive WTD tested for SARS-CoV-2 compared to contemporaneously collected CWD-negative WTD. CWD column indicates the groups that are CWD positive or CWD negative. The counts positive and percent positive columns indicate the counts and percent SARS-CoV-2 positive for each CWD group.**

| CWD | Counts | Counts Positive | Percent Positive |
|---|---|---|---|
| Negative | 408 | 44 | 10.78% |
| Positive | 196 | 16 | 8.16% |

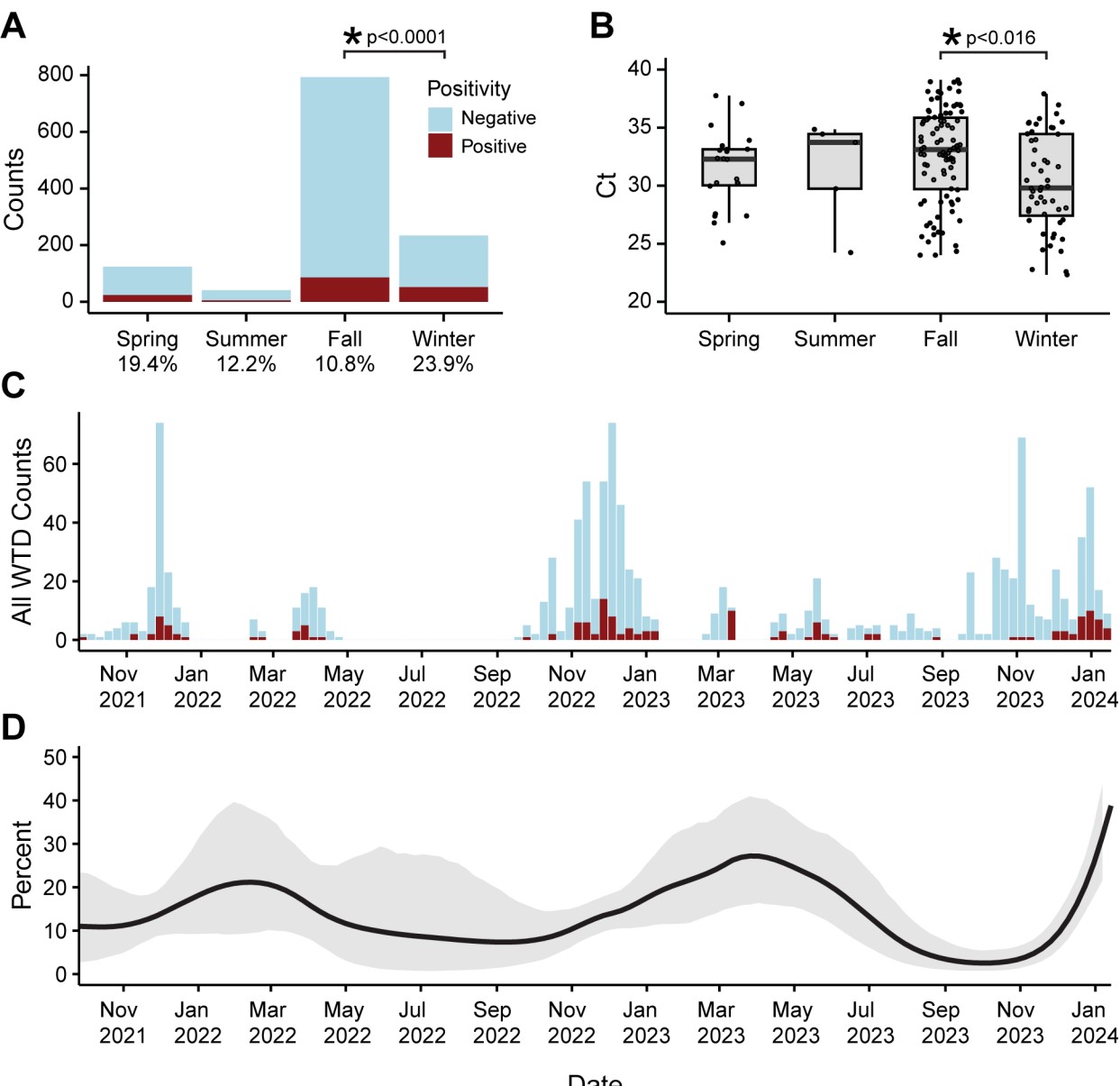

**Fig 2. RT-qPCR for pooled WTD from Pennsylvania 2021–2024 analyzed by season.** (A) RT-qPCR counts stratified by season (x-axis: spring, summer, fall, and winter). Blue indicates SARS-CoV-2 negative, red indicates SARS-CoV-2 positive by RT-qPCR. The y-axis represents the number of counts in each season. (B) Ct values of RT-qPCR positive samples stratified by season. (C) Prevalence in aggregate data from RPLN and nasal swabs binned by week. Blue indicates SARS-CoV-2 negative and red indicates SARS-CoV-2 positive by RT-qPCR. The y-axis represents the number of counts per week; x-axis indicates time. (D) Auto-regressive moving average logistic regression estimating SARS-CoV-2 positivity by RT-qPCR over time. Grey indicates the 95% credible interval. The y-axis represents the estimated percent positive of the sampled population; x-axis indicates time.

with temperatures below 10°C (n = 824 WTD). There was no statistically significant difference (two-sample test for equality of proportions with continuity correction, comparing samples collected above and below 10°C (p = 0.25)).

To model positivity over time in our data, we used a Bayesian autoregressive moving average logistic regression to smooth data and estimate SARS-CoV-2 positivity rates in

Pennsylvania WTD (Fig 2C and 2D). We observed several relative maxima that correlate with higher prevalence in winter and spring compared to fall and summer. We estimate average weekly SARS-CoV-2 positivity rates in the fall to be 9.8% (95% CrI 5.4% – 15.7%), and summer 7.5% (95% CrI 2.0% – 18.6%) compared to winter 21.9% (95% CrI 12.2% – 34.7%) and spring 17.5% (95% CrI 8.7% – 31.1%).

## Geographic spread of SARS-CoV-2 in WTD 2021–2024

To estimate positivity rates across different land use types, we used publicly available GIS data to determine the most common land use for each 5 km radius around each sampled deer location [50] (Fig 3A and 3B). We found that deer in predominantly forested areas (13.16% positive, n = 623) were significantly less likely to be positive than deer in crop and pasture areas (21.21% positive, n = 132) (p = 0.024, Chi-Squared). This suggests that proximity to human activity is associated with increased SARS-CoV-2 prevalence in WTD. Moran's I was implemented to measure spatial autocorrelation, which refers to the correlation of a signal among nearby spaces [51]. Analysis indicated that the degree of landscape fragmentation surrounding the WTD was not significantly different between positive and negative animals (p > 0.48, Wilcox test).

To determine the extent of clustering of SARS-CoV-2 infections in WTD, we calculated the joint-count statistic at varying spatial scales. This statistic assesses the frequency of positive-positive pairs for the binned time periods. Using a permutation test to determine enrichment, we found no significant clustering. No trend was found for positivity by county when stratified by year 2021–2024 (S3 Fig).

## Characterization of the geographical distribution of viral variants via viral spike gene sequencing

To characterize the viral variants in WTD, we extracted RNA from positive samples and performed viral-targeted sequencing. Initial amplicon sequencing was performed using a nested primer set targeting the SARS-CoV-2 spike receptor binding domain (RBD) coding sequence, as previously described [45,52]. *42* of the 165 RT-qPCR positive samples yielded spike RBD sequences. RT-qPCR Ct strongly correlated with recovering enough sequence data to call variants with a linear trend predicting probability of sequencing attributed to Ct having a p < 0.0001. There was a 51% probability (95% CI 44% – 59%) that samples with Ct values of 23 yielded viral sequences, and a 12% probability (95% CI 5% – 18%) that sample with Ct values of 40 yielded viral sequences.

We identified 17 Alpha, 11 Delta, and 14 Omicron WTD-derived isolates. We found Alpha and Delta variants more than a year after they were last reported as the major circulating variants in humans, extending previous findings of protracted circulation in WTD (Fig 4A–4C) [45,53]. We found that Alpha variants were observed clustered in WTD from the eastern half of the state and Delta variants clustered in the western half of the state (Fig 4D).

## Variation within lymph nodes

To assess within-lymph node variability, we collected multiple samples from single lymph nodes and performed RT-qPCR. The seven locations sampled included 1) exudate from the area surrounding RPLN samples, 2) cranial medial portion of the RPLN, 3) center of the RPLN, 4) distal medial portion of the RPLN, 5) cranial lateral portion of the RPLN, 6) efferent vessel, and 7) adipose tissue removed from the surface of the RPLN (S4A Fig). We performed RT-qPCR across two replicates of each location for 8 WTD lymph nodes (S4B Fig). Seven of these were previously identified as SARS-CoV-2 positive and one as SARS-CoV-2 negative.

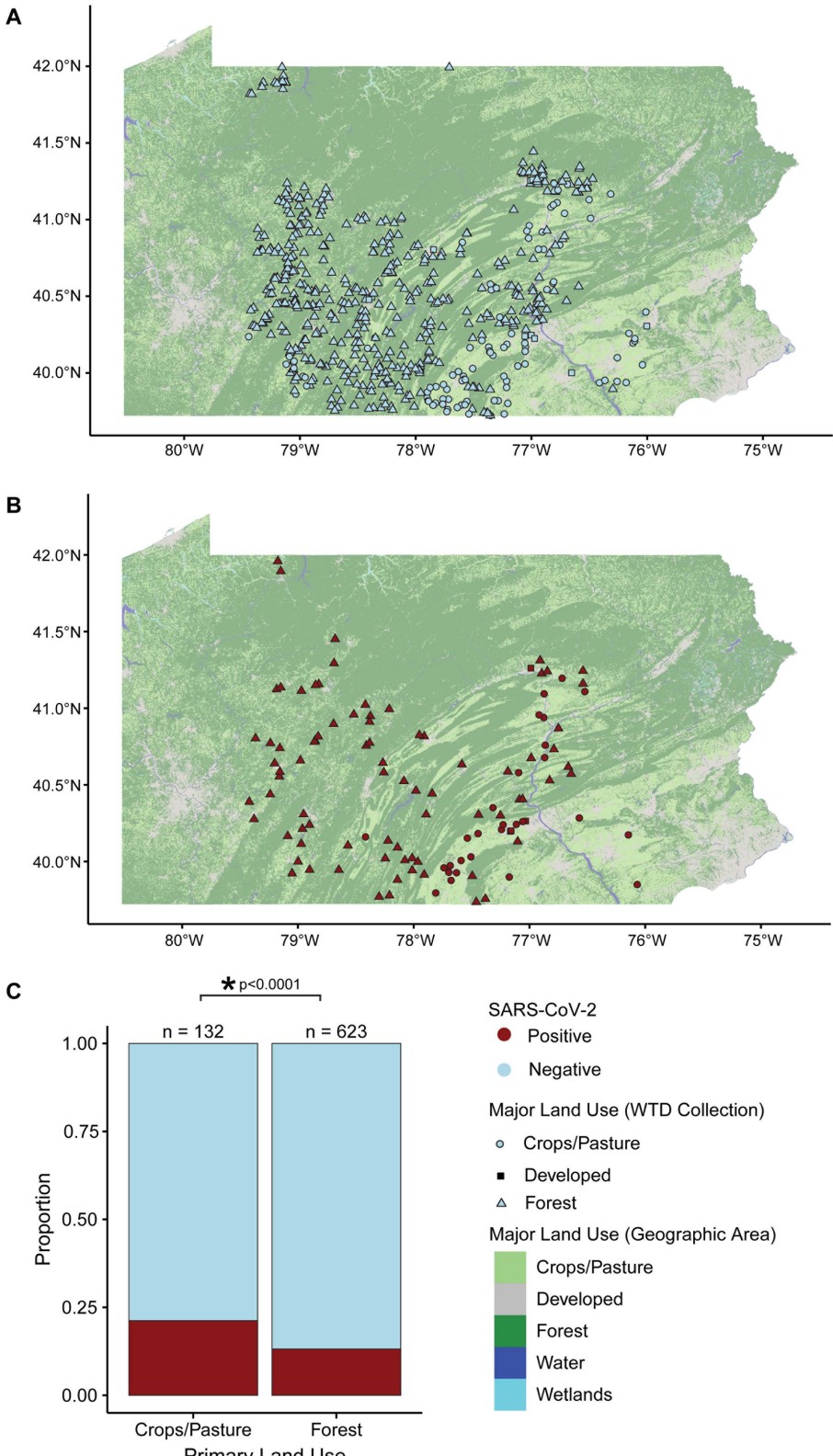

**Fig 3. Land use and clustering analysis of SARS-CoV-2 in WTD in Pennsylvania.** (A and B) Maps of Pennsylvania colored by land use category. The x and y-axes show latitude and longitude. The color of the data point indicates whether the sample was positive (red) or negative (blue) for SARS-CoV-2 by RT-qPCR. The shape of the data point

indicates the major land use category from the area that the WTD was collected. Circles indicate crops/pasture, squares indicate developed, and triangles indicates forested. (A) Locations of SARS-CoV-2 positive WTD. (B) Locations of SARS-CoV-2 negative WTD. (C) Stacked bar plot showing the proportion SARS-CoV-2 positive (red) and negative (blue) WTD from predominantly crops/pasture compared to forested areas. The y-axis shows the proportions, the x-axis the land use category.

We hypothesized that regions that were taken as residual exudate or adipose surrounding the RPLN would be positive with a lower viral RNA level than the samples cut from the RPLN, but variability would be greatest when comparing deer-to-deer verses within deer. We hypothesized that single variants would be found within each lymph node, and not divergent genomes from multiple different infections. We found a greater amount of Ct variation across RPLN locations compared to variation within WTD (standard deviation of WTD: 2.74, standard deviation of sampling RPLN location 3.32). To determine if there was a significant difference across RPLN locations, we performed a two-way ANOVA, and found no significant difference among sites (p = 0.93). In contrast, there was a significant difference when comparing lymph node specimens from different WTD (p = 0.013). We also performed a Bayesian regression analysis to estimate the effect size of samples and location on Ct values. Similarly, we found a significant effect from deer-to-deer sampling (S4C Fig) but no significant effect from sampling location within the RPLN (S4D Fig).

To compare viral variants found in different RPLN locations, three samples were analyzed by amplification and sequencing of spike RBD coding regions (position 22,773 to 23,323), including three different viral variants (VSP30363 (Delta), VSP30426 (Alpha), and VSP30917 (Omicron)). We hypothesized that different RPLN locations would harbor similar lineages, all derived from a single infection. Not all sites produced amplicons, but those that did clustered tightly with other sequences from the same samples (S4E Fig and S4F Fig). There was one instance of a sample harboring a single nucleotide difference (sample VSP30917) with position 22814 called as a G, a U, or no coverage. However, this sequence is at the start of the amplicon, where quality of sequence reads is lowest, and so is of uncertain significance. All other substitutions were identical throughout the RBD of this sample, and within RBD of all other samples. Thus the lymph node sequences each identified infections with only a single viral variant.

To test whether biopsy samples from the core of lymph nodes showed different results than samples from superficial tissues, we collected eight WTD samples known to be SARS-CoV-2 positive and performed RT-qPCR in replicates for the core versus surface cuts (S4G Fig). We found that biopsied core samples have significantly higher Ct values than superficial cuts (p = 0.0035, two-tailed paired t-test). We implemented a Bayesian multivariate regression analysis to control for variation across samples, which confirmed that cut depth had a significant association with Ct, with surface cuts having a Ct difference of -1.73 (95% CrI 0.80 to 2.65) compared to the biopsy cut. This change represents a 3.25 fold increase (95% CI 1.74 to 6.28) in RNA copies for surface vs biopsy cuts.

## Analysis of viral whole genome sequences

Twenty-five whole genome sequences were recovered from the RNA samples that yielded the original 42 spike RBD amplicon sequences. For analysis, we contextualized these WTD-derived isolates by comparison to a set of human-derived isolates from our region and across the USA.

In these data, we identified 7 likely spillover events from human-to-deer. Criteria for identifying events as independent included phylogenetic branches that were separated by human-derived isolates. We did not detect any close matches of human and deer-specific lineages that would suggest deer-to-human spillback (Fig 5A and S3 Table). We then pooled our

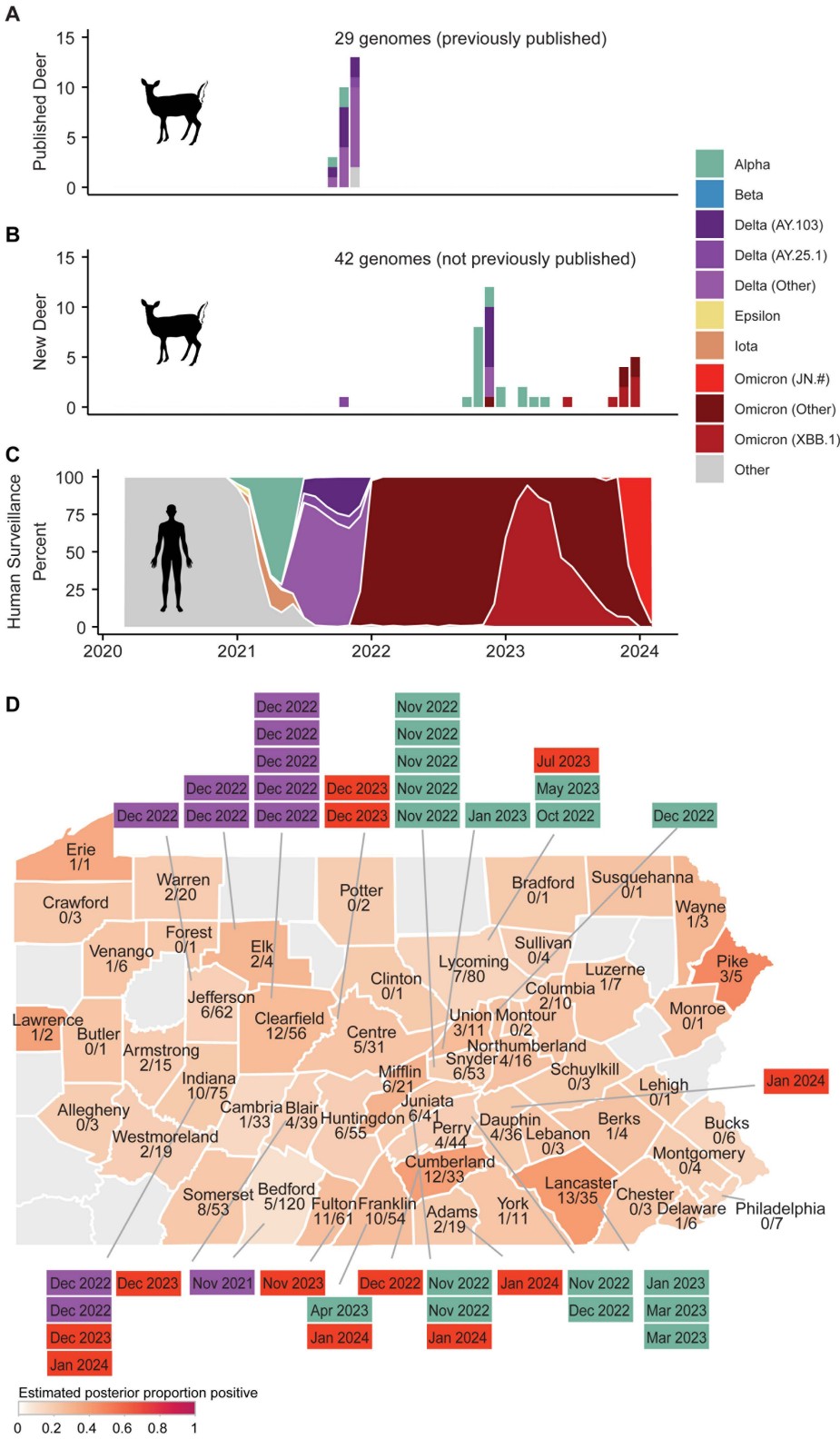

**Fig 4. Circulating SARS-CoV-2 lineages in Pennsylvania WTD.** (A) Previously published WTD-derived SARS-CoV-2 sequences from Pennsylvania, with annotated county-level geographic information[45]. The x-axis represents time from 2020 to 2024, the y-axis represents the counts of each lineage, and the color indicates the identified variant/lineage from the deer-derived isolate. (B) WTD-derived SARS-CoV-2 sequences contributed from this paper

(collection spanning 2021–2024). Markings as in (A). (C) Human-derived isolates subsampled to 10,000 Pennsylvania sequences from January 2020 to February 2024. The x-axis indicates time from 2020 to 2024, the y-axis represents the percent of total sequences for a given month from the subsampled 10,000 sequences available. Colors indicate the annotated viral variants (key to right). (D) Map of Pennsylvania showing ratio of samples testing positive for SARS-CoV-2, colored by estimated posterior proportion positive by county, with boxed labels containing month of collection and colored by SARS-CoV-2 variant. Teal indicates Alpha, purple indicates Delta, and red indicates Omicron.

newly sequenced WTD samples with previously published data from Pennsylvania WTD with > 98% coverage from GISAID, which documented at least 12 total spillovers from humans to WTD in Pennsylvania since the beginning of the epidemic. With our limited dataset, we estimate approximately 1 spillover event every 68 days (collection 10/22/2021 to 1/12/2024) (S5A Fig).

We identified Alpha and Delta variants persisting in WTD for months to years after they had largely ceased to circulate in humans. This suggests long-term onward spread of SARS-CoV-2 in the WTD population, establishing WTD as a potential reservoir for ancestral lineages. Experimental infections suggest that individual WTD likely clear their SARS-CoV-2 infection within days or weeks [54,55]. If true of the deer sampled here, the data suggest potential transmission from deer-to-deer, with short-term infection of each. Moreover, the Alpha strains that persisted in deer were as divergent from their nearest human-derived viral sequences as Omicron was from Alpha (Fig 5B–5C). The least divergent WTD Alpha lineage was 72 mutations from its nearest human-derived sequence, data consistent with deer-to-deer spread, or possibly spread involving unsampled intermediate animal species.

The most recent isolates from WTD were found to be exclusively Omicron, marking a shift in circulating lineages within Pennsylvania WTD. This recent increase in WTD positivity for winter 2024 shows what appear to be four independent XBB.1 spillovers from humans into deer, coinciding with an increase in human cases of this variant. However, the more recent JN.1 lineage of the Omicron variant, which had largely driven the spike of SARS-CoV-2 human cases at that same time, has not been identified in Pennsylvania WTD samples to date. This could be due to factors we were unable to test such as possibly increased capacity for XBB.1 to infect WTD. The absence of Alpha and Delta variants in the most recent samples could be attributed to a small sample size. However, this coincides with a decline in positivity rate followed by a rapid increase in positivity rate once the Omicron XBB.1 lineages emerged in WTD. This pattern is consistent with a shift in circulating lineages.

An analysis of evolution rates within variants suggested that there was no significant difference in rates between human-circulating variants compared to deer-circulating variants (p = 0.94). We then asked if there were greater than expected mutations in deer isolates when accounting for substitutions that might emerge during the original introduction to deer from humans. To test this we used a permutation test to determine whether deer samples were more likely than randomly selected human samples to be outliers from the historic evolution rate. This test found a non-significant trend with a probability of 0.0523. Together, there is no evidence given this limited data that jumping from human-to-deer resulted in a higher genetic divergence rate in WTD than that observed in humans. Further, once circulating in deer, the virus did not diverge at a significantly faster rate than observed in human-to-human transmission.

We performed time-resolved phylogenetic analyses using BEAST to estimate the time to the most recent common ancestor for our 7 spillover events (S5 Fig). Alpha cluster A was estimated to have its last common shared human ancestor on 4/1/2021 (95% highest posterior density (HPD) 2/5/2021 to 4/15/2021). This would suggest persistence of this Alpha lineage within deer for 1.76 years (95% CrI 1.72-1.91 years) which is a minimum estimate as

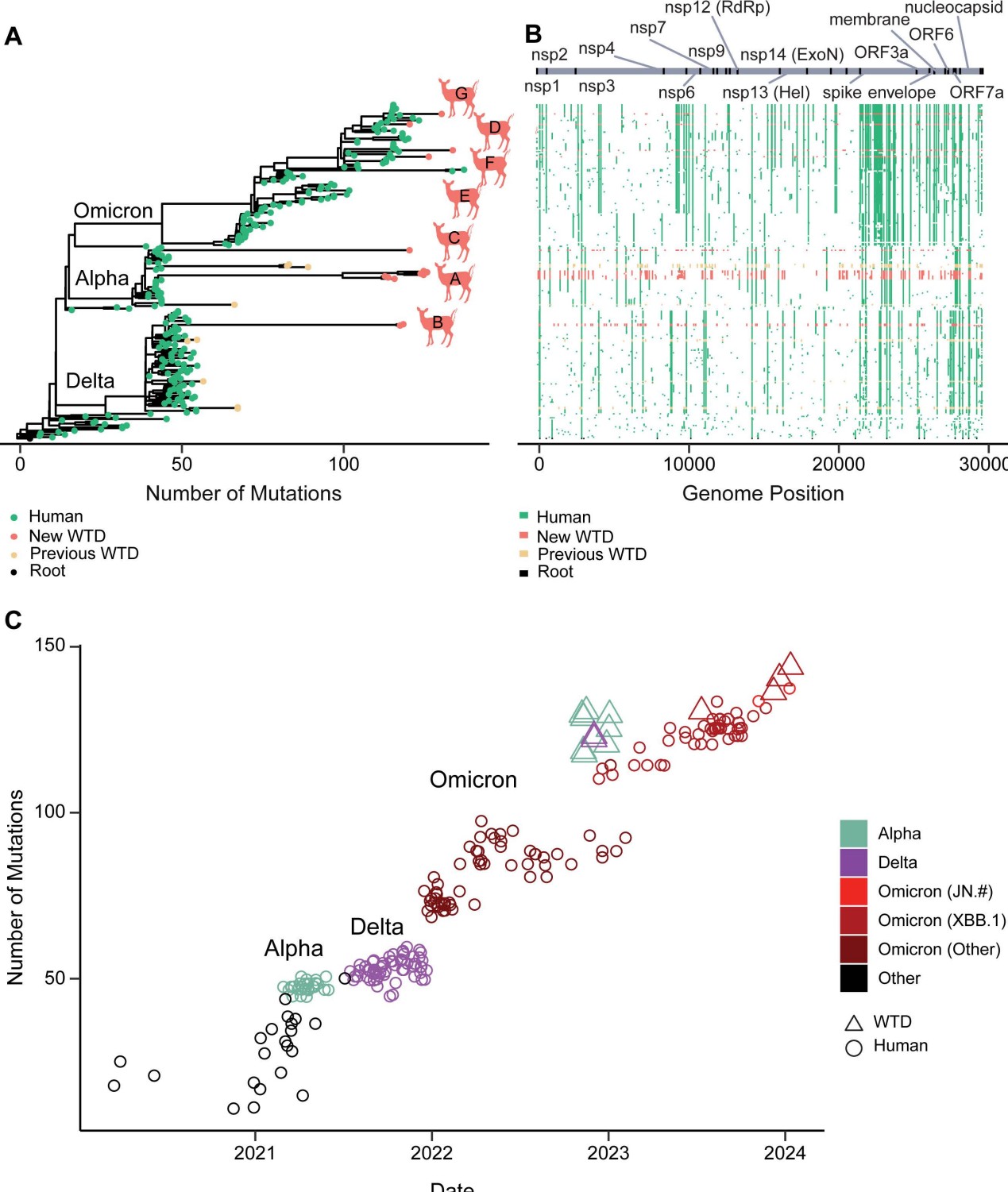

**Fig 5. Phylodynamics of Pennsylvania WTD SARS-CoV-2 genomes.** (A) The seven newly sequenced WTD SARS-CoV-2 isolates on a phylogenetic tree also containing subsampled human-derived isolates spanning the course of the SARS-CoV-2 pandemic. Genomes are labeled by lineage, and colored by host species. Green indicates human-derived isolates, light red indicates WTD-derived isolates new to this paper, and dark red indicates WTD-derived isolates from previous work [44,45,53]. Labeled letters indicate the suspected seven independent spillover events documented in this work. (B) Mutation heatmap paired with the phylogenetic tree showing the location of SNPs throughout the SARS-CoV-2 genome, colored by host species. Mutations are marked in reference to the earliest uploaded sequence to public databases (NC_045512.2). The x-axis represents the

position in the SARS-CoV-2 genome and the y-axis represents an individual sample, paired with the data points on the phylogenetic tree. (C) Root-to-tip plot showing genetic divergence as the number of mutations from first introduction to the United States, colored by SARS-CoV-2 lineage with open circles indicating WTD-derived isolates and closed circles indicating human-derived isolates. The y-axis represents the time that the sample was collected and the x-axis represents the number of mutations accumulated in reference to the first detection of SARS-CoV-2 in the USA.

it assumes that the spillover occurred at the time of the last common shared ancestor and not before. The analysis also assumes that the spillover was directly from humans to deer without an intermediate host. Alpha cluster C was estimated to have its last common ancestor with humans on 3/14/2021 (95% HPD 2/15/2021 to 3/22/2021), suggesting potential circulation in deer for 1.81 years (95% CrI 1.78-1.88 years).

The Delta variant also appeared to persist in WTD. Cluster B was estimated to share its last common ancestor on 5/28/2021 (95% HPD 3/27/2021 to 8/15/2021), corresponding to 1.52 years (95% HPD 1.30-1.69 years) of persistence in deer.

The XBB.1 Omicron lineage had an estimated 4 independent spillover events with their most recent common ancestor dating to 4/17/2023 (95% HPD 3/14/2023 to 5/5/2023), 8/19/2023 (95% HPD 6/11/2023 to 9/19/2023), 8/7/2023 (95% HPD 6/22/2023 to 8/29/2023), and 7/15/2023 (95% HPD 5/24/2023 to 7/31/2023). This gives a minimum estimated persistence of 0.24 years (95% CrI 0.19 – 0.33 years), 0.30 years (95% CrI 0.22 – 0.49 years), 0.37 years (95% CrI 0.31-0.50 years), and 0.50 years (95% CrI 0.45-0.64 years).

To examine regions in the SARS-CoV-2 genome that had increased numbers of mutations, we used the nearest human isolate for each of the WTD as a reference to mark substitutions that likely newly arose in WTD (Fig 6A). We performed a Poisson regression analysis using total counts in each coding region as the response and protein product as the predictor with an offset of log(length). We observed several coding regions to have more mutations than expected (Fig 6A) including nsp4 (p = 0.0027), nsp6 (p = 0.018), nsp7 (p = 0.037), nsp15 (p = 0.024), ORF3a (p = 0.012), ORF7a (p = 0.021), ORF7b (p = 0.0021), and ORF8 (p = 0.0045).

To determine if substitutions emerged independently in multiple lineages, we again use the nearest human-derived isolates to determine which substitutions could have arisen in WTD. Of the substitutions we previously found enriched in WTD isolates, C7303U, C9430U, and C20259U [45], we observed them to again emerge independently 6, 3, and 3 times for the WTD isolates new to this study. Among other mutations that appeared to emerge through multiple independent WTD clusters, we observed spike G252V (common in XBB.1 and its sublineages [56]) revert to G252 in 4 independent clusters. These 4 independent reversions encompass all clusters of the XBB.1 lineage (or its sublineages). Thus in all cases of XBB.1 spillover we observed a loss of spike G252V. These emergences all appeared as reversions from U (XBB.1) to G (Wuhan) substitution in nucleotide position 22,317. This is the strongest signature of WTD-specific selection from this dataset. The only other nonsynonymous mutation observed to emerge independently in 3 or more cases was nsp4 S3149F (encoded by a C9711U nucleotide substitution). The importance of this substitution is unknown.

To assess which elements within the viral genome were under the most selective pressure, we performed a dN/dS analysis over SARS-CoV-2 coding regions for each WTD whole genome sequence recovered (Fig 6B). We found differences that were not statistically significant when comparing dN/dS values among deer (Kruskal-Wallis p = 0.053). However, there was a significant difference among coding regions (Kruskal-Wallis p < 0.0001). Non-structural protein coding regions were under the greatest diversifying selection, including nsp9, ORF6, and ORF7a. This contrasts with human-derived isolates where structural components including spike, nucleocapsid, and envelope were among the genes under the strongest diversifying selective pressure [37,57–59].

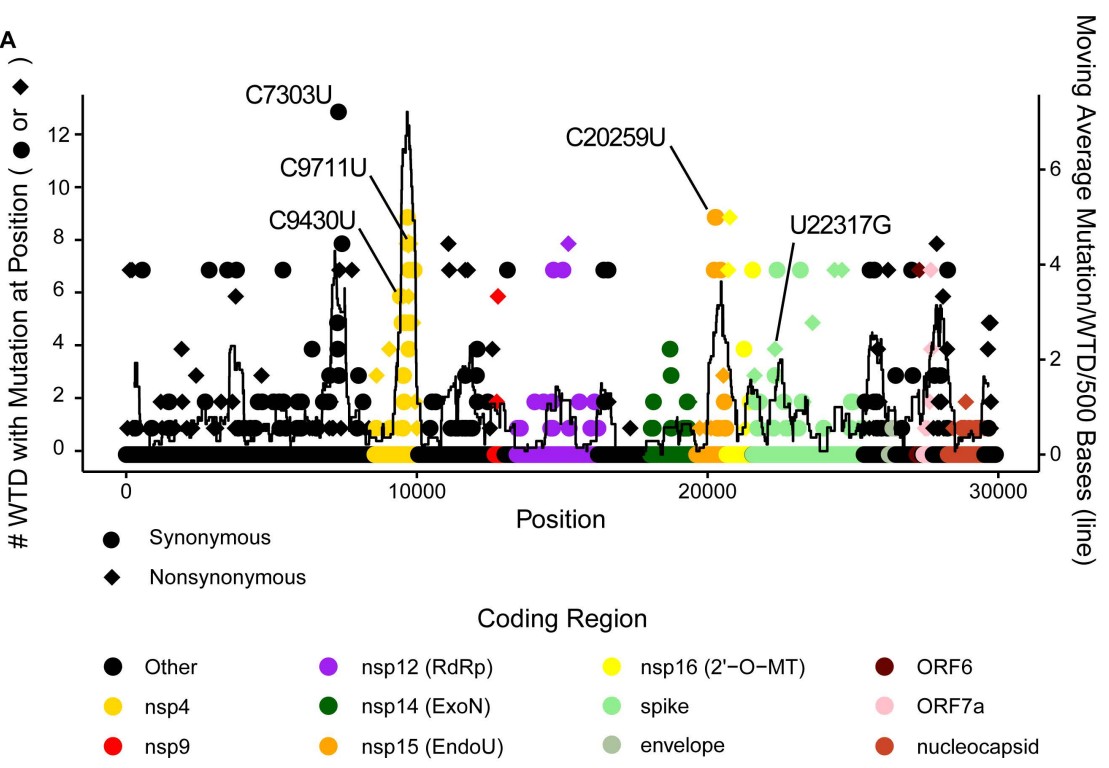

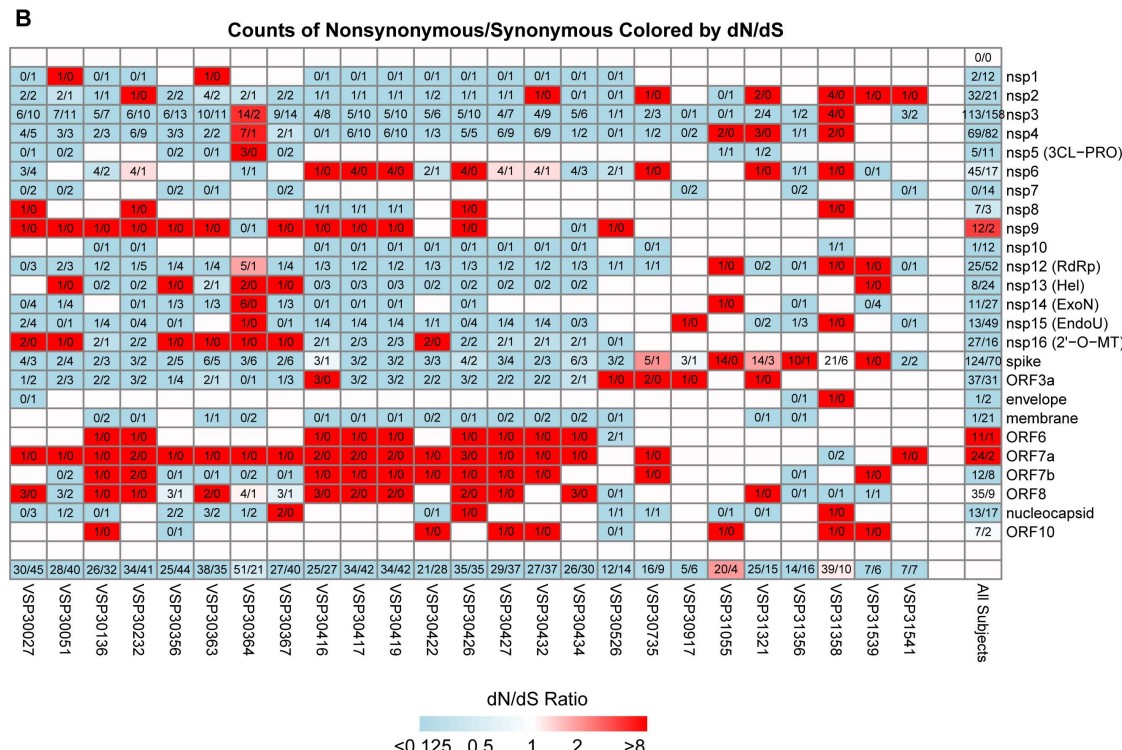

**Fig 6. Analysis of accumulation of mutations.** (A) Plot of WTD SARS-CoV-2 mutation occurrence in reference to nearest human neighbor. Mutations for each WTD were determined using their nearest genetic human isolate as a reference. Circles indicate synonymous substitutions and diamonds represent nonsynonymous substitutions. Colors indicate the corresponding coding region. The data points (circle or diamond) represent how many WTD had a mutation at this position in the genome. The

lines represent the moving average number of mutations per WTD per 500 base pair window. The y-axis on the left corresponds to the data points (circle or diamond) and the y-axis on the right corresponds to the moving average black line. (B) Nonsynonymous and synonymous mutation analysis (dN/dS analysis). The figure shows a heatmap for SARS-CoV-2 coding regions and WTD-derived isolates with labeled ratios for counts of nonsynonymous substitutions to synonymous substitutions colored by dN/dS ratio. Counts indicated as 4/7 would represent 4 nonsynonymous substitutions with 7 synonymous substitutions. Colors indicate the dN/dS ratio that is derived from counts of nonsynonymous divided by total possible nonsynonymous divided by counts of synonymous divided total possible synonymous. The darkest shade of blue indicates a dN/dS ratio of 0.125 or lower and the darkest shade of red indicates a dN/dS ratio of 8 or higher. Blue indicates that there are more than expected synonymous substitutions compared to nonsynonymous substitutions. Red indicates there are more than expected nonsynonymous substitutions than synonymous substitutions. Along the bottom row is a summary for each coding region in the WTD sampled (all coding regions within an individual animal). Along the right side of the grid is the summary for a coding region across all WTD (all isolates for a coding region). "VSP" indicates "viral specimen number".

## Discussion

Here we report new geographic, temporal, and genomic data on the spread of SARS-CoV-2 in Pennsylvania WTD from 2021 to 2024. Our data provide evidence for seven further spillovers of SARS-CoV-2 from humans to deer in Pennsylvania; together with previous data, the results now support twelve independent spillovers to date [44,45,53]. No clear examples were seen of spill back from WTD to humans in our data. The basis of spillover from human to deer is unclear. We identified land use associated with crops and pasture as having a higher association with WTD prevalence than forest, suggesting increased prevalence associated with proximity to humans. Our data also identified a significant seasonal trend, with the highest infection rates in winter and spring. We found no differences in infection rates or viral RNA load associated with the prion disease CWD. Genomic analysis identified WTD as a reservoir for ancestral Alpha and Delta variants, which persisted in WTD for more than 1.5 years after transmission from humans. We did find multiple examples of the same mutations appearing independently in different WTD SARS-CoV-2 spillovers, documenting apparent selective pressures unique to WTD.

In our previous study of WTD in Pennsylvania, we identified three silent substitutions that recurred in WTD, C7303U, C9430U, and C20259U [45] --remarkably, these were seen in the new transmission events 6, 3, and 3 times from unique spillovers new to this paper. In addition, a characteristic XBB.1 spike mutation G252V was observed to revert to G252 in all four independent XBB.1 spillovers. Review of publicly available WTD-isolates returned just one example of an XBB.1 virus (collected November of 2023 in Minnesota), and this virus did not have the reversion to ancestral G252 (GenBank PP692379.1). A preprint describing WTD in Texas also identified XBB.1 sublineages, however none of these had the spike G252V reversion to G252 [60]. These data provide new evidence for potentially WTD-specific evolutionary trajectories and pose novel questions on mechanisms of WTD immune responses

Several coding regions were identified as having more mutations than expected including nsp4, nsp7, nsp15, ORF3a, ORF7a, ORF7b, and ORF8. nsp4 is important for SARS-CoV-2 replication and the formation of reticulovesicular networks. nsp6 is thought to restrict autophagosome expansion [61]. nsp7 is involved in creating an nsp7-nsp8 complex to aid with nsp12 polymerase activity [62]. nsp15 contains an EndoU domain required for RNA processing [63]. ORF3a is thought to be one of three ion channels [64]. ORF7a interacts with structural proteins M, E, and S and contains a transmembrane domain [65]. ORF7b acts to inhibit the innate immune defense mechanism's RLR signaling pathway [66]. ORF8 is an accessory protein thought to interfere with immune responses [67]. None were structural proteins; several were implicated in modulating immune responses suggesting a possible basis for selection.

Seasonality of SARS-CoV-2 in WTD may be driven by 1) behavioral changes when deer aggregate in larger groups during the more resource-limited colder months, 2) aggregation

associated with mating, or 3) environmental changes with colder-drier air allowing more efficient transmission of viral particles from deer-to-deer [68]. Seasonality of other coronaviruses is well established in humans including coronaviruses 229E, OC43, and NL63 [69]. Seasonal trends for SARS-CoV-2 in humans have previously been reported, although it is still not clear how much of this is driven by climate-related changes, changes in host behavior, or the timing of emergence of new variants [70], adding interest to understanding WTD as a model. In Pennsylvania, WTD breed from mid-October to mid-December, with a peak in mid-November [71]. Winter and early spring are also seasons when deer aggregate in larger groups, something that occurs less in the late spring as deer separate in part to avoid attracting predators [72]. These seasonal gatherings of larger groups could allow for more efficient deer-to-deer transmission.

Our genomic data document long-term persistence and genetic divergence of SARS-CoV-2 Alpha and Delta in the WTD population more than a year after circulation ended in humans. Such long-term evolution has the potential to generate new variants insensitive to human immune responses. The WTD reservoir could also potentially sustain a viral lineage outside of a largely immune/vaccinated human population long enough for human immunity to wane, enabling spill-back to humans and re-emergence of ancestral lineages [73].

Evolution rates of SARS-CoV-2 in free-range WTD have previously been reported to be up to three times higher than those observed in humans [74], but our study did not find elevated rates. Our analysis focused on the post-establishment phase of SARS-CoV-2 in WTD, comparing the first WTD sample to subsequent WTD chains of transmission in the same lineage. These transmissions occurred 18 months or more after the estimated initial spillover. In contrast, a previous study assessed rates 8 months or fewer after estimated spillover and potentially captured accelerated evolution that could be attributed to adaptation to a new host species [74]. Additionally, the previous study assumed no unsampled intermediate hosts between human and WTD, which could have potentially contributed to differences in observed rates. Thus our study and previous work are not directly comparable, and so not contradictory.

Analysis of co-infected WTD with SARS-CoV-2 and CWD has not been previously published, and here we report no association in the population sampled. The lack of association suggests that the behavioral changes, physiologic burden, or malnutrition associated with CWD does not influence animal transmission of SARS-CoV-2. However, information on disease progression in animal samples was not available. Thus whether there are differences between early CWD versus disease-progressed animals remains to be determined.

The mode of SARS-CoV-2 spillover from humans to WTD is not known. Experimental evidence suggests that infectious particles can persist in aerosols for from 4.8 minutes to up to 2 hours depending on conditions of sunlight, humidity, and temperature [75,76]. This could potentially be sufficient for aerosol transmission between species if they were in proximity. Urbanization may increase the human-deer geographical intersect which has been shown to be associated with increased disease prevalence in some deer species in more developed regions [74,77,78]. Transmission to WTD through contaminated water has been suggested, but wastewater has yet to be identified as source of SARS-CoV-2 transmission for any organism [79,80]. It is also suggested that transmission from humans could occur through fomites or other infected animal species [81].

This study has several limitations. Insufficient genomic surveillance of circulating SARS-CoV-2 strains in humans more recently might cause us to miss examples of spill back from deer to humans. Genomic background sequences were selected following a subsampling scheme for most samples with the inclusion of most-similar publicly available sequences, so not all strains of possible relevance were compared. Sample size was a limitation in some of the statistical analyses. Samples were not available for immunological profiling. Many samples were collected for CWD testing, and therefore were stored at ambient temperature or 4°C for hours to days before being transferred to -20°C and then

-80°C for long term storage, likely contributing to variability in Ct values and ability to recover sequence data.

Our findings, and those of others, have broad public health implications. This study underscores the need to further investigate the routes of spillover and spill-back transmission, which remain incompletely defined. The persistence of SARS-CoV-2 in WTD and the emergence of genetically divergent isolates in these animals signals a potential risk of reintroduction of both novel and re-emergent viral strains into humans.

## Methods

### Samples and tissues

This study did not involve live animals. Diagnostic samples from deceased animals were submitted to the Pennsylvania Animal Diagnostic Laboratory System at the New Bolton Center for CWD testing. The residual tissues were donated for this research.

No human participants were involved in the data collection for this study. Data referencing human-derived viral isolates were obtained from previously published online databases. See the data availability section and S3 Table for additional information.

### Sample collection

Retropharyngeal lymph node (RPLN) samples from free-ranging white-tailed deer were submitted to the Wildlife Futures Program (WFP) at the Pennsylvania Animal Diagnostic Laboratory System – New Bolton Center (PADLS-NBC) for CWD diagnostic testing. Samples were primarily collected from hunter harvested deer and road killed deer. Fresh samples had various times from collection to receipt by the lab and may have been at ambient temperatures for several days but generally reached the lab within one week. Once received by the lab, samples were stored at 4°C (1-5 days) until initial CWD testing. All samples were tested using BioRad's TeSeETM Short Assay Protocol (SAP) Combi Kit ELISA for CWD. RPLNs with confirmed negative or positive CWD status were then stored at –20°C until they were used in this study.

### RT-qPCR

RPLN tissue was trimmed for RT-qPCR analysis and placed in a Precellys 2 mL Hard Tissue Homogenizing Ceramic Bead Tube (REF: P000911-LYSK1-A) containing 500 µL of PBS. If needed the tube was inverted to ensure the sample was immersed in PBS prior to homogenization. Once the samples were added to their homogenization tubes, positive and negative extraction controls were generated. The negative extraction control was a blank containing only PBS, and the positive extraction control was PBS with cell-derived viral RNA spiked in from an in-house SARS-CoV-2 RNA stock (gift from Dr. Susan Weiss, strain WA-1). After the controls were generated, all samples were homogenized. Crucially, the samples had to be sufficiently homogenized to allow for testing, but not homogenized so thoroughly that they would obstruct the RNA extraction columns used in subsequent steps. Samples were homogenized at 18 Hz for 4 minutes in the Qiagen TissueLyser II.

RNA extraction for RT-qPCR was carried out using the Qiagen RNeasy Mini Kit (REF: 74106) as previously described [45,49]. Two regions of the viral nucleocapsid gene, nucleocapsid 1 (N1) and nucleocapsid 2 (N2), were targeted. CDC N1 Probe and Primers (REF: 10006713) were used for the detection of nucleocapsid 1, and CDC N2 Probe and Primers (REF: 10006713) were used to detect nucleocapsid 2. Both probe and primer sets were used with TaqMan Fast Virus 1-Step Mix (REF: 4444434). Reactions were performed with a 50°C step at 5 minutes, 95°C for 20 seconds, then 45 cycles of the following: 95°C for 3 second, 60°C for 30 seconds. Each extracted RNA sample was tested for N1 and N2 in duplicate, and any sample where three

of four RT-qPCR tests were positive was considered SARS-CoV-2 positive. A negative control from a blank RNA extraction, and a negative control from a no template qPCR were used. Two positive controls were generated by spiking in synthetic RNA (one positive control) and cell culture-derived viral RNA (a second positive control) at the RNA extraction steps. The synthetic RNA was IDT SARS-CoV-2 Plasmid DNA (REF: 10006625) and the cell-derived viral RNA was from an in-house RNA stock (a gift of the Weiss lab, strain WA-1). If either of the negative controls amplified by 45 cycles, then the data from the run was invalidated.

## Autoregressive moving average (positivity over continuous time)

To estimate smoothed positivity rates over time, we implemented a Bayesian autoregressive moving average model using Markov Chain Monte Carlo (MCMC) sampling in STAN [82]. This estimates positivity rate for a given week that regresses to the average of the previous week and the rate of change from one week should be similar to rate of change from the previous week.

Equation 1

$$P_w = \frac{e^{X_w}}{\sum_{j=1}^{n} e^{X_{j,w}}}$$

$$X_w = X_{w-1} + change_w$$

$$change_w \sim Normal(change_{w-1}, \sigma)$$

Where $P_w$ is the true positivity rate for week $w$. $change_w$ is the change in log odds for week $w$. The standard deviation, $\sigma$, had a prior gamma distribution with shape parameter of 1 and scale parameter of 2. $X_w$ had the flat prior given by a normal distribution with mean of 0 and standard deviation of 10.

## Logistic regression (positivity over multiple variables)

To estimate the effect of SARS-CoV-2 seasonality in WTD we used a Bayesian regression model for estimating the effect size of fixed effect variables for the season of infection, average weekly temperature of that region of Pennsylvania during time of sample collection, and animal sex. We define the region as the nearest NOAA weather observation station with average temperature recorded. We also included a random effect for cause of death. Parameters were estimated using an MCMC method.

Equation 2

$$\text{logit}(P(Positivity = 1)) = \beta_0 + \beta_1 Sex + \beta_2 Season + \beta_3 Temperature + u_{Death_i}$$

Where $P$(Positivity = 1) represents the probability of a given sample testing SARS-CoV-2 positive, logit($P$) is the log-odds of the probability $P$, $\beta_0$ represents the intercept, $\beta_1$ represents the effect for the season, $\beta_2$ represents the effect of male sex on SARS-CoV-2 positivity, $\beta_3$ represents the effect of average regional temperature during sample collection on SARS-CoV-2 positivity, and $u_{Death_i}$ represents the random effect for the cause of death on animal's probability of SARS-CoV-2 positivity.

## Spike-targeted amplicon sequencing

Amplicon sequencing targeted the spike coding region using nested PCR. Extracted RNA underwent reverse transcription and initial DNA amplification using the Superscript

IV One-Step RT-PCR System. The reaction mixture consisted of 5 μL of extracted viral RNA, 0.25 μL of Superscript IV, 12.5 μL of 2X Platinum SuperFi RT-PCR mix, 1.25 μL of each primer at 10 μM concentration, and 4.75 μL of molecular grade water, totaling 25 μL. The cycling conditions were set to: 2 minutes at 25°C, 20 minutes at 50°C, 2 minutes at 95°C, followed by 25 cycles of 95°C for 2 minutes, 55°C for 30 seconds, and 70°C for 1 minute. The first PCR reaction amplified spike RBD from position 22732 to 23403 (NC_045512.2) using the following primers: CTGCTTTACTAATGTCTATGCAGATTC and TCCTGATAAAGAACAGCAACCT.

The second PCR, a nested reaction, added sequencing adapters. It included 5 μL of the product from the first reaction, 12.5 μL of 2X Q5 hot start master mix, 0.5 μL of dNTPs (10 mM), 1.25 μL of each primer (10 μM), and 7.5 μL of water, also totaling 25 μL. The conditions were 95°C for 2 minutes, followed by 20 cycles of 95°C for 15 seconds, 55°C for 30 seconds, and 72°C for 1 minute. The second PCR reaction amplified a nested region from position 22773 to 23322 and adds sequencing adapters. The second PCR reaction is denoted as follows where lowercase values indicate sequence adapters and uppercase indicates homologous regions to SARS-CoV-2 genome: tcgtcggcagcgtcagatgtgtataagagacagGTGAT-GAAGTCAGACAAATCGC and gtctcgtgggctcggagatgtgtataagagacagATGTCAAGAATCT-CAAGTGTCTG. These sequencing adapters allow sequencing to occur either on Illumina's platform after barcoding, or direct Sanger sequencing.

The PCR product was visualized using a 1% agarose gel to determine whether any fragments of 550 base pairs were produced. Samples that had expected amplification band sizes were sent for Sanger sequencing at the UPENN sequencing core.

## Whole genome sequencing

To sequence viral genomes, the POLAR protocol was used. Initially, 5 μL of RNA sample was combined with several reagents: 0.5 μL of Random Hexamers, 0.5 μL of a dNTPs mix (10 mM), and 1 μL of nuclease-free water. This mixture was heated at 65°C for 5 minutes. For reverse transcription, 6.5 μL of this heated mixture was added to 0.5 μL of SuperScript III Reverse Transcriptase, 2 μL of First-Strand Buffer, 0.5 μL of DTT (0.1 mM), and 0.5 μL of RNaseOut, and then incubated at 42°C for 50 minutes followed by 10 minutes at 70°C.

For the amplification of cDNA, 2.5 μL of the transcription reaction was mixed with 0.25 μL of Q5 Hot Start DNA Polymerase, 5 μL of Q5 Reaction Buffer, 0.5 μL of a dNTPs mix (10 mM), and ARTIC primers for separate pools. (4.0 μL for the first set, 3.98 μL for the second set), with additional water to bring the total volume to 25 μL. The samples underwent 25 cycles of heating to 98°C for 30 seconds, cooling to 65°C for 5 minutes, then briefly heated again to 98°C.

Afterward, the two reactions were combined, and the DNA was prepared for sequencing using the Nextera XT Library Preparation Kit. Libraries were tagged using IDT DNA/RNA UD Indexes and quantified with the Quant-iT PicoGreen dsDNA assay. Libraries were sequenced on the Illumina MiniSeq platform.

## Phylogenetic analysis

To investigate the relationship of WTD SARS-CoV-2 variants to those in humans, we recovered 25 whole genome sequences from the 42 samples studied as spike RBD amplicons. BLAST was used to determine the 5 most similar (nearest neighbor) sequences from the 5,025,365 available sequences from the USA on GISAID downloaded 3/25/2024 (S3 Table). Augur was then used to subsample 50 human sequences from anywhere in the USA and 100 additional sequences from Pennsylvania and its neighboring states, New York, New Jersey,

Maryland, Delaware, West Virginia, and Ohio. Subsampling was done within each lineage identified in our WTD sequence data. These pooled sequences were then aligned using NextClade and IQ-Tree was used to construct a maximum-likelihood phylogenetic tree. These subsampled isolates were used to contextualize the WTD whole genome sequences recovered. Phylogenetic analysis was performed on samples downloaded from GISAID, subsampled using Augur v24.3.0, aligned using Nextclade v2.14.0, and maximum-likelihood trees inferred using IQ-TREE v1.6.12 with 1,000 bootstrap replicates.

For lineage-specific analyses, the root was the earliest detected that isolate sampled in the United States with complete date information and > 99% coverage. For multi-lineage analyses, the same criteria was used for root selection without restricting for lineage type. Previously reported WTD and their 5 nearest neighbors that had > 98% coverage were included. Trees were visualized using ggtree (an R package) in the provided code.

### d N/dS analysis

dN/dS was used to estimate if a given coding region was undergoing neutral evolution, positive selection, or purifying selection. Analysis was implemented by using the following set of equations:

Equation 3

$$dN = \frac{N}{N_{site}}$$

$$dS = \frac{S}{S_{site}}$$

$$dN/dS = \frac{dN}{dS}$$

Where N represents the number of nonsynonymous substitutions detected and $N_{site}$ represents the counts of all possible nonsynonymous substitutions that could occur. S represents the number of synonymous substitutions detected and $S_{site}$ represents the counts of all possible synonymous substitutions that could occur. Substitutions detected for N and S were calculated by using each WTD-derived isolate's most genetically similar human-derived isolate using the BLAST method described previously.

### Time-resolved phylogenetic analysis

A time-scaled Bayesian maximum clade credibility (MCC) tree was used to estimate the introduction of SARS-CoV-2 lineages into Pennsylvania WTD. We selected genomes based on their closest human neighbors using GISAID's database limiting results to isolates collected in the USA, resulting in a pool of 5,025,365 SARS-CoV-2 genomes from which the nearest 5 sequences were pulled using BLASTn. If a deer's closest matches included another deer, both were analyzed in the same tree. Sequences were filtered to have 98% coverage and exact collection dates were required. Subsampling was performed using Nextclade Augur tool on sequences downloaded from GISAID. We subsampled to include 100 human-derived isolates from the same lineage restricted to the sampled region (Pennsylvania, New York, New Jersey, Delaware, West Virginia, Maryland, and Ohio). The trees were rooted with the earliest detection of the lineage that had > 99% coverage and complete collection data. An additional 50 sequences subsampled from anywhere in the USA was used. Each dataset was aligned using NextClade with Wuhan-Hu-1 as

a reference (NC_045512.2). An MCMC method in BEAST v1.10.4 was used to generate Bayesian molecular clocks[83]. A general time reversible substitution model with gamma-distributed rate variation and an uncorrelated relaxed lognormal clock was implemented with a Bayesian Skyline tree with a group size of 10 [44,45,84–86]. MCMC sampling ran for 100 million iterations, with every thousandth iteration sampled and 10 million iterations discarded as burn-in. We report our HPD which is the region of the posterior probability distribution which is the smallest possible interval that contains the 95% probability mass under the posterior interval. BEAGLE 3 improved computational performance. Tracer v1.7.1 assessed convergence, TreeAnnotator v1.10.4 summarized the MCC tree, and FigTree v1.4.4 visualized it [83,87]. All tree analyses were repeated three times independently to confirm convergence and effective sample sizes above 200. Maximum-likelihood trees were generated by IQ-TREE v1.6.12.

## RPLN subsampling study

To examine the distribution of viral RNA levels and genomic diversity present within single RPLN samples, harvests from multiple locations on several specimen were conducted. Seven RPLNs were chosen for this experiment, six known RT-qPCT positives and one RT-qPCR negative included as a control. Sample locations included exudate and adipose tissue surrounding the RPLN and sections from cranial medial, distal medial, cranial lateral, efferent vessel, and center regions of the RPLN sample. For each sampling location, two pieces of tissue were harvested following the protocol above and processed independently. RT-qPCR and spike-targeted sequencing was performed as described above.

## Geospatial analysis

Land use data were downloaded from the Pennsylvania Spatial Data Access (PASDA) portal (https://www.pasda.psu.edu/uci/DataSummary.aspx?dataset=2514). Land cover was determined from the 2021 Pennsylvania Cropland raster dataset at 30-meter resolution. All crops and pasture were collapsed into a single land use, as were all forests, all wetlands, and all developed land categories. The terra (https://cran.r-project.org/web/packages/terra/index.html) and raster (https://cran.r-project.org/web/packages/raster/index.html) R packages were used for raster data processing, and vector data was processed with the sf package (https://cran.r-project.org/web/packages/sf/index.html). Join-count statistics were calculated manually in R.

## Supporting information

**S1 Fig. Comparison of SARS-CoV-2 positivity over time stratified by CWD status.** (A) CWD positive WTD are shown over the sampling period. Deer numbers are shown as stacked bar plots (y-axis) with blue indicating SARS-CoV-2 negative and red indicating SARS-CoV-2 positive by RT-qPCR. Time is shown on the x-axis. (B) CWD negative WTD sampled over time. Markings as in A with blue indicating SARS-CoV-2 negative and red indicating SARS-CoV-2 positive by RT-qPCR. The y-axis represents the counts. Red indicates SARS-CoV-2 positive and blue indicates SARS-CoV-2 negative by RT-qPCR. The x-axis labels apply to both panels.
(TIF)

**S2 Fig. RT-qPCR positivity over time and weekly temperature at time of collection.** Each data point represents an individual WTD, colored by SARS-CoV-2 positivity, with blue indicating SARS-CoV-2 negative and red indicating SARS-CoV-2 positive by RT-qPCR. The y-axis represents temperature in Celsius averaged over the week of collection (x-axis) for the region that the sample was collected in.
(TIF)

**S3 Fig. Pennsylvania SARS-CoV-2 positivity in WTD by year from 2021 to 2024.** (A-D) Maps of Pennsylvania, with counties indicated. Each country is annotated for the ratio of positive SARS-CoV-2 WTD number of WTD tested. Grey indicates a county that had no tests performed that year. White indicates a county had tests performed and there were no positives. Shades of red indicate the proportion of SARS-CoV-2 positive specimens. Data are presented separately for (A) 2021 (B) 2022 (C) 2023, and (D) 2024.
(TIF)

**S4 Fig. Consistency of SARS-CoV-2 sequences and viral RNA levels in different locations within each RPLN.** (A) Graphical illustration of WTD indicating RPLN locations of the seven sample types, including exudate, cranial medial, center, distal medial, cranial lateral, efferent, and adipose tissue. Seven WTD previously identified as SARS-CoV-2 positive were tested and one WTD previously identified as negative. (B) RT-qPCR Ct values stratified by sample and colored by RPLN site. Each dot represents the average of two replicates. Undetermined RT-qPCR results are indicated as points positioned at the top of the plot. "VSP" indicates "viral specimen number" (accession number). (C) Posterior distribution for Ct effect size of each sample (VSP) from a Bayesian linear mixed model. A value of 0 indicates that the variable had no effect on Ct value. (D) Posterior distribution for Ct effect size of each RPLN site. (E) Phylogenetic tree of spike-targeted sequencing results from three WTD, emphasizing within-animal consistency. (F) Paired with (E), map showing SNP location in the sequenced region of the SARS-CoV-2 spike, colored by sample, with grey indicating low coverage. (G) Ct value comparison of biopsied core vs superficial cuts across 8 WTD specimen with each dot representing a single replicate. The y-axis shows the Ct value; the x-axis shows the lymph node sampled. The bar represents an average across replicates.
(TIF)

**S5 Fig. Time-resolved viral phylogenetic trees based on SARS-CoV-2 whole genome sequences, comparing WTD isolates to human isolates.** For all trees, colors reflect the host organism with green indicating human-derived isolates and pink indicating WTD-derived isolates. The icon of a WTD includes a letter specifying the cluster that the WTD belonged (Fig 5A-5G). (A) Time-resolved phylogeny of Alpha variant (B.1.1.7) WTD including nearest-neighbor sequences and subsampled background human sequences. (B) Time-resolved phylogeny of Delta variant (AY.103). (C) Time-resolved phylogeny of Omicron variant (XBB.1).
(TIF)

**S1 Table. Animal species reported to be infectable with SARS-CoV-2.** Only species with well documented infections fsrom scientific articles (peer review or preprint) were consider credible reports of animal infection.
(CSV)

**S2 Table. Sample metadata including WTD demographics, date of collection, cause of death, county of death, and sequence accession.** Negative is abbreviated as neg, and positive is abbreviated as pos. n1_average_ct and n2_average_ct are the average Ct across the two replicates if they were both positive. n1_half_positive and n2_half_positive are the Ct values for samples where one of the two replicates yielded a detectable Ct. Samples were only declared positive if three of four tests were positive. hict_lineage is the lineage as called by the nested RBD sequencing approach. Values in hict_lineage will either be the lineage called or the step at which the nested RBD sequencing approach failed. hict_length is the number of nucleotides recovered from the nested RBD sequencing. wgs_lineage is the lineage called from whole genome sequencing attempt. wgs_percent_5x_coverage is the percent of the

SARS-CoV-2 genome that had 5x coverage or greater for whole genome sequencing. wgs_VSP is the viral specimen accession assigned to the sequence data. seq_variant is the variant call from whole genome sequencing. seq_cluster is the designated phylogenetic cluster to which the sequence belonged. ncbi_accession is the GenBank accession that can be used to download the sequence. gisaid_accession is the GISAID accession that can be used to download the sequence. Isolate is the isolate name for uploading the sequence. CWD is the chronic wasting disease status for the deer as either positive (pos) or negative (neg). CWDLabel1 is the accession for the CWD test. IncidentDate is the date that the animal was initially collected. Age is the age of the WTD sampled. Gender specifies male or female for the WTD. County-Value specified the county that the sample was collected in. Region specifies the region that the sample was collected in using cardinal directions within the state of Pennsylvania. Latitude and Longitude specify the coordinates that the sample was found. Animalstatus specifies the reported cause of death for the WTD.
(CSV)

**S3 Table. Samples and accession numbers for sequence data used in this study.** This includes accession using GenBank and/or GISIAD accessions for background samples, phylogenetic roots, and nearest neighbor sequences.
(CSV)

## Acknowledgements

We are grateful to members of the Bushman, Gagne, and Collman laboratories for help and suggestions. We thank Laurie Zimmerman for help with artwork and manuscript preparation. We thank the numerous staff with the Pennsylvania Game Commission for the collection of the RPLN samples as well as Jan Yacabucci and John Armstrong for conducting the CWD diagnostic testing.

## Author contributions

**Conceptualization:** Andrew D. Marques, Matthew Hogenauer, Natalie Bauer, Roderick B. Gagne, Frederic D. Bushman.

**Data curation:** Andrew D. Marques, Matthew Hogenauer, Natalie Bauer, Michelle Gibison, Beatrice DeMarco, Roderick B. Gagne.

**Formal analysis:** Andrew D. Marques, Matthew Hogenauer, Beatrice DeMarco, Scott Sherrill-Mix, Carter Merenstein, Roderick B. Gagne.

**Funding acquisition:** Ronald G. Collman, Roderick B. Gagne, Frederic D. Bushman.

**Investigation:** Andrew D. Marques, Matthew Hogenauer, Natalie Bauer, Beatrice DeMarco, Scott Sherrill-Mix, Carter Merenstein, Ronald G. Collman, Roderick B. Gagne, Frederic D. Bushman.

**Methodology:** Andrew D. Marques, Matthew Hogenauer, Natalie Bauer, Beatrice DeMarco, Scott Sherrill-Mix, Roderick B. Gagne, Frederic D. Bushman.

**Project administration:** Andrew D. Marques, Natalie Bauer, Michelle Gibison, Roderick B. Gagne, Frederic D. Bushman.

**Resources:** Andrew D. Marques, Michelle Gibison, Roderick B. Gagne, Frederic D. Bushman.

**Software:** Andrew D. Marques, Scott Sherrill-Mix, Carter Merenstein.

**Supervision:** Andrew D. Marques, Michelle Gibison, Scott Sherrill-Mix, Carter Merenstein, Roderick B. Gagne, Frederic D. Bushman.

**Validation:** Andrew D. Marques, Matthew Hogenauer, Scott Sherrill-Mix, Carter Merenstein, Roderick B. Gagne.

**Visualization:** Andrew D. Marques, Matthew Hogenauer, Carter Merenstein.

**Writing – original draft:** Andrew D. Marques, Roderick B. Gagne, Frederic D. Bushman.

**Writing – review & editing:** Andrew D. Marques, Natalie Bauer, Michelle Gibison, Scott Sherrill-Mix, Carter Merenstein, Ronald G. Collman, Roderick B. Gagne, Frederic D. Bushman.

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
