## [Decision Letter · Decision Letter 0]

13 Sep 2024

Dear Dr. Bushman,

Thank you very much for submitting your manuscript "Evolution of SARS-CoV-2 in White-tailed Deer in Pennsylvania 2021-2024" for consideration at PLOS Pathogens. As with all papers reviewed by the journal, your manuscript was reviewed by members of the editorial board and by several independent reviewers. In light of the reviews (below this email), we would like to invite the resubmission of a significantly-revised version that takes into account the reviewers' comments.

We cannot make any decision about publication until we have seen the revised manuscript and your response to the reviewers' comments. Your revised manuscript may be sent to reviewers for further evaluation.

Sincerely,

Shuo Su

Academic Editor

PLOS Pathogens

Ronald Swanstrom

Section Editor

PLOS Pathogens

Michael Malim

Editor-in-Chief

PLOS Pathogens

orcid.org/0000-0002-7699-2064

Reviewer's Responses to Questions

**Part I - Summary**

Reviewer #1: In this study the authors sampled deceased white-tailed deer in Pennsylvania for the presence of SARS-CoV-2 which has been reported as a possible reservoir in this study and previous studies. They reported isolation of RNA from retropharyngeal lymph nodes and performed qRTPCR which was positive for 14% of the samples. They did whole genome sequencing of the 25 samples and conducted detailed sequence and phylogenetic analysis which revealed the circulation of Alpha, Delta, and Omicron variants of SAR-Cov-2 among WTD. They concluded that there is no correlation between a prion disease and SAR-Cov-2 positivity. The results are based on collected samples from 2021 to 2024 in forested regions as well as regions closer to human activities where majority of the positive samples were identified. The presence of the virus in WTD samples were reported to peak in Spring and Winter which could be related to deer behavior or human behavior and peak of human infections as well. Moreover, the variants which were circulating among humans were identified in WTD simultaneously, but also Alpha and Beta variants were reported to be present in deer samples additional one year after losing dominance in human samples. Their phylogenetic studies show several spillover events of different variants from human to deer populations not vice versa and occurrence of high number of mutations after the spillover. Although these results support the hypothesis of WTD being a reservoir of the virus, the recent isolates don’t show the presence of Alpha or Beta variants.

WTD being a possible reservoir of SAR-Cov-2 was also pointed in previous studies by other groups and the authors of this study. For this reason, this study is important as a follow up, providing new analysis. I think to study such an impactful human pandemic virus in One Health approach is very important since the SARS-CoV-2 is still circulating and mutating among humans and other animals. Although the sample number is not too high, this study provides detailed genomic and phylogenetic analysis of the variants circulating in WTD in Pennsylvania in 2021-2024. Sampling other animals living in the proximity to WTD and evaluating how the change in environment and climate effects the virus circulation would be a nice One Health approach for future studies.

Reviewer #2: This study focused on the genetic characterization of SARS-CoV-2 in White-tailed deer (WTD; Odocoileus virginianus) in Pennsylvania, USA from 2021 until 2024, including the spatial and temporal occurrences of these infections. Many animal species have been shown to be zoonotic reservoirs for SARS-CoV-2, including WTD, some of which bear public health significance due to the close contact with human populations, facilitating spillover between species. Deer-to-deer and potential human-to-deer or deer-to-human transmissions need to be studied in order to determine SARS-CoV-2 circulation patterns, the implications on SARS-CoV-2 evolution and the emergence of new variants, and potential future spillover events. These data contribute to the emerging picture of SARS-CoV-2 onward transmission and can inform infection control management.

Reviewer #3: This study includes a comprehensive testing of lymph node samples removed from white-tailed deer (WTD) in the state of Pennsylvania, USA, during 2021-2024. The samples were tested for the presence of SARS-CoV-2 RNA, and positive samples were further studied for genomic classification. Different variables of the population such as age, geographic location, sex, infection of chronic wasting disease (CWD) and other characteristics, were studied with respect to SARS-CoV-2 infection. The study involves molecular biology methods (PCR, RNA sequencing), several bioinformatics analyses and various statistical analyses. This is a comprehensive study, and it is novel in the sense that it examines the WTD population in the studied area, as a reservoir for SARS-CoV-2 and as a possible source for animal-to-human spillback source. The study also demonstrates that SARS-CoV-2 variants that disappeared from human population were still circulating among WTD population months later.

Reviewer #4: In this submission, the authors present a nicely executed study looking at the evolution of SARS-CoV-2 in white-tailed deer (WTD) populations coupling efficiently, routine surveillance for another disease (CWD) affecting deer, with molecular, epidemiological and ecological analyses on the circulation of the virus. The sampling is of decent size and spans a period of almost 3 years which, to my knowledge, is worth noting to be the longest period described for a given study on SARS-CoV-2 in wildlife. Therefore, the dataset generated and analyses performed in this work have the potential to satisfy the PLOS Pathogens readership. However, while ecological aspects such as correlations (or not) of SARS-CoV-2 prevalence and viral loads with age, sex, cause of death, seasons and land use are comprehensively addressed the molecular evolutionary part on the other hand seems more superficial. Indeed, the data show strong signs of extended circulation of different SARS-CoV-2 variants (e.g Alpha and Delta) but fail to present clearly what are the mutational signatures at the protein level. This study kind of sits in between a description of the spatio-temporal circulation of SARS-CoV-2 in WTD populations from one single state in the US and the potential for a detailed analysis of the viral evolution in WTD and how that compares to what’s described in the human population. Altogether, a deeper investigation around these aspects is of strong interest and could benefit the community broadly.

**Part II – Major Issues: Key Experiments Required for Acceptance**

Reviewer #1: (No Response)

Reviewer #2: None

Reviewer #3: The molecular analyses – real-time PCR, Sanger sequencing and Illumina genomic sequencing, are key techniques that were crucial for obtaining the data. Subsequently, the statistical and bioinformatic methodologies that were used, such as applying Bayesian regression model to test connection between different study parameters, using Time-Resolved Phylogenetic Analysis for genomic sequences, were essential for the data analysis in this study. The combined use of these approaches enabled interpretation and inference of the collected data.

Reviewer #4: - A substantial proportion of samples present Ct values ≤35. To which extend can we talk about positivity resulting from a productive infection vs any potential environmental or experimental contamination?

- Figure 5: I might be wrong, but this figure does not seem to be called in the text when its result are presented.

It would be really interesting to present aa mutational profiles alongside genomic location of SNPs. The reader might find himself left with a bitter taste in the mouth after expecting to find a mine of information which could rapidly be translated into further investigations in the community.

A focus on spike mutational profiles looks obvious given its important in driving immune evasion, tissue tropism and virulence and since the authors targeted the RBD region but also interesting to present for others such as nsp6, ORF6, and N. Comparative approaches with what’s described in the human population would be beneficial, if possible. Comparing with Omicron BA.1 and BA.2.86 (both very likely resulting from saltation events over the course of chronic infection or local circulation within a given population) could be meaningful to potentially identify patterns or hotspots of mutations regardless of the species.

- lines 318-324, Do the results really support the conclusion suggested by the authors? Are the conditions well controlled to discriminate between recent versus ancient/long term infections? Do we have the information of the percentage of animals having such highly mutated sequences? In other words, could we affirm that the most highly mutated sequences do spread/circulate in WTD population or is it coming from a very small percentage of samples individuals? Could we expect chronic/long infection in WTD?

- Line 342-343 and likely Figure 5: the authors conclude that “the virus did not diverge at significantly faster rate than observed in human-to-human transmission”. This is different from McBride et al. Nature Communications 2023 where they estimated the evolution rate to be 3 times faster in WTD compared to human. And I’m surprised that previous study is not cited and discussed in the manuscript. What could explain this difference in conclusions reached?

**Part III – Minor Issues: Editorial and Data Presentation Modifications**

Reviewer #1: Line 56: I think it would be more informative if the authors can add the geographical information to the Supp Tables 1. For ex, a column of the countries where the animals were captured.

Line 94: How do the authors think a prion disease could potentially trigger a new SARS-CoV-2 variant? Since the readers are likely naïve to the prion disease in WTD, it would be helpful to explain how it effects the general health of deer and if it weakens the immune system so that they are more prone to other infections such as SAR-Cov-2. Also, the authors should mention here if deer show any SARS-CoV-2 symptoms in their study and in previous reports?

Line 104: Please provide the meanings of abbreviations used in the Supp Table 2 where applicable to make understanding clear for wider readers. It would also be useful to explain terms such as “n1_half_positive”, “hict_lineage” etc. These can be added somewhere in the Table or header.

Line 151: Please mention the red represents the SARS-CoV-2 positive samples and the blue represents the negative ones.

Line 275: Please cite the previous publications.

Figure 5: This figure is not cited in the Text! What is the reference to calculate the Mutation numbers? Can you please provide the acc. number of the reference in the text? I wanted to note that the provided resolution of the Figure 5 didn’t let me to see the colors and lineages clearly, especially for Fig5A.

Line 283: Did the authors identify recurring amino acid substitutions among the WTD isolates?

Line 320: The lack of previous isolates in the recent samples contradict the idea of the WTD being a reservoir. Do you think that it is due to the number of samples?

Line 305: Please indicate the acc. Numbers of the isolates used to create the phylogenetic in Figure 5A. I think Table S2 must be cited. Please clarify in Table S3 which sequences are from WTD.

Lines 309-310: Please site the previous works.

Line 442: Please cite the “previous data”.

Line 445: I think spillover from humans to WTD begs more discussion. What could be the route of transmission between WTD and humans? This can be about human behavior. How often humans and WTD occupy the same environment? What kind of activities humans do in WTD occupied regions in the sampled places? Do humans get in close contact with WTD or for how long SAR-CoV-2 can remain in the air according to the published studies?

Line 451: If WTD became a reservoir for Alpha and Delta variant, why more recent isolates don’t have these variants?

Line 608: “primers specific to the study” please be more specific and cite.

Reviewer #2: Throughout paper: Suggest switching to standard date format “yyyy/mm/dd”.

Line 300: replace “out” with “our”.

Line 519 and 536: replace “doped in/doping in” with “spiked in/spiking in”.

Line 595: Please clarify what is meant by “no notable SARS-CoV-2 sequences were detected” when results for the sequencing are presented on p13.

Ref 63 has been published already – please amend (doi: 10.1371/journal.pone.0294283)

Line 650: “MCMC” defined in this line, but the abbreviation is not always used in the text (where the full name is used; e.g. line 544).

Reviewer #3: The order in which the figure legends are placed within the main text is confusing and makes it difficult to follow the flow of the manuscript. I suggest that the main text will be organized as continuous flow of paragraphs, and the figure legends will be placed subsequently. Of course, in the finalized version, the legends will be placed next to the figures, so this should be easier to follow.

My other comment regarding the legends, is the need to provide more detailed explanations regarding the different panels in each figures, to improve clarity. As many of the figures are complex and contain several panels, a more detailed explanation could be helpful.

Lastly, I think that placing the methods section before the results makes more sense, but the Editors and Journal policy dictate this.

Reviewer #4: - The sample distribution in Fig.S3 seems to be showing predictive effect of temperature on positivity with apparently a higher abundance of positive samples at temperatures ≤ 10°C despite the text stating it did not. Could the authors double check the analysis?

- lines 215-222, how does this correlate with prevalence in human according to the different seasons?

- Figure 5: Use different colours between new WTD sequences and those from previous work. It is difficult to distinguish between both groups otherwise.

- Line 328, Could this be due to XBB.1 lineage having gained/increased capacity to infect deer?

- Line 389-436 and Figure 6: I am not entirely sure what to think about this section and associated results. It sounds more like nice supplementary data and findings which were performed as control rather than being critical for the study and conclusions. Especially if placed in front of the mutational signatures observed in WTD compared to human which lack in this manuscript, as mentioned before.

- Line 445, “how this is happening so efficiently…” maybe ton down that statement as it is not clear yet how efficient this is versus rare spillovers followed by efficient spread once in WTD populations.

- Line 455-470, some statements and hypotheses are popping without results directly linked to them and analysis of correlation are missing.

PLOS authors have the option to publish the peer review history of their article (what does this mean? ). If published, this will include your full peer review and any attached files.

**Do you want your identity to be public for this peer review?** For information about this choice, including consent withdrawal, please see our Privacy Policy .

Reviewer #1: No

Reviewer #2: No

Reviewer #3: No

Reviewer #4: No
---

## [Decision Letter · Decision Letter 1]

18 Dec 2024

PPATHOGENS-D-24-01513R1

Evolution of SARS-CoV-2 in White-tailed Deer in Pennsylvania 2021-2024

PLOS Pathogens

Dear Dr. Bushman,

Thank you for submitting your manuscript to PLOS Pathogens. After careful consideration, we feel that it has merit but does not fully meet PLOS Pathogens's publication criteria as it currently stands. Therefore, we invite you to submit a revised version of the manuscript that addresses the points raised during the review process.

This is clearly an important study. The only concern voiced by one reviewer has to do with accessibility to the reader. We will not offer any specific advice but rather ask you to consider changes you feel are appropriate to improve accessibility of the story to a broader audience. We believe this exercise will be to everyone's benefit, including the authors as these changes could enhance the impact of the work.

Please submit your revised manuscript within 30 days Feb 16 2025 11:59PM. If you will need more time than this to complete your revisions, please reply to this message or contact the journal office at plospathogens@plos.org. Please include the following items when submitting your revised manuscript:

We look forward to receiving your revised manuscript.

Kind regards,

Shuo Su

Academic Editor

PLOS Pathogens

Ronald Swanstrom

Section Editor

PLOS Pathogens

Sumita Bhaduri-McIntosh

Editor-in-Chief

PLOS Pathogens

orcid.org/0000-0003-2946-9497

Michael Malim

Editor-in-Chief

PLOS Pathogens

orcid.org/0000-0002-7699-2064

**Journal Requirements:**

1) We have noticed that you have uploaded Supporting Information files, but you have not included a list of legends. Please add a full list of legends for your Supporting Information files after the references list.

2) Please ensure that the funders and grant numbers match between the Financial Disclosure field and the Funding Information tab in your submission form. Note that the funders must be provided in the same order in both places as well.

**Reviewers' Comments:**

Reviewer's Responses to Questions

**Part I - Summary**

Reviewer #1: To my opinion, the authors fairly addressed the reviewer comments. I believe, this study presents results which could be useful for further, more detailed analysis concerning SARS-Cov-2 reservoir and circulation among WTD in Pennsylvania and elsewhere.

Reviewer #2: This study focused on the genetic characterization of SARS-CoV-2 in White-tailed deer (WTD; Odocoileus virginianus) in Pennsylvania, USA from 2021 until 2024, including the spatial and temporal occurrences of these infections. Many animal species have been shown to be zoonotic reservoirs for SARS-CoV-2, including WTD, some of which bear public health significance due to the close contact with human populations, facilitating spillover between species. Deer-to-deer and potential human-to-deer or deer-to-human transmissions need to be studied in order to determine SARS-CoV-2 circulation patterns, the implications on SARS-CoV-2 evolution and the emergence of new variants, and potential future spillover events. These data contribute to the emerging picture of SARS-CoV-2 onward transmission and can inform infection control management.

The authors strengthen their findings from a previous publication by increasing sampling over a 3 year period and show evidence of seven SARS-CoV-2 spillovers from humans to WTD, either directly or through an unknown intermediary. They show no evidence for faster evolution of sequences in white-tailed deer, although whole-genome recovery from infected WTD samples was at a very low frequency. CWD status of the WTD does not affect SARS-CoV-2 incidence in this study. The analysis into potential evolutions hotspots within SARS-CoV-2 genomes from WTD is interesting.

Limitations: the authors state that storage conditions of the samples prior to

Reviewer #3: This study includes a comprehensive testing of lymph node samples removed from white-tailed deer (WTD) in the state of Pennsylvania, USA, during 2021-2024. The samples were tested for the presence of SARS-CoV-2 RNA, and positive samples were further studied for genomic classification. Different variables of the population such as age, geographic location, sex, infection of chronic wasting disease (CWD) and other characteristics, were studied with respect to SARS-CoV-2 infection. The study involves molecular biology methods (PCR, RNA sequencing), several bioinformatics analyses and various statistical analyses. This is a comprehensive study, and it is novel in the sense that it examines the WTD population in the studied area, as a reservoir for SARS-CoV-2 and as a possible source for animal-to-human spillback source. The study also demonstrates that SARS-CoV-2 variants that disappeared from human population were still circulating among WTD population months later.

Reviewer #4: First of all, I would like to thank the authors for providing a substantially revised version of the manuscript according to reviewers' comments. The authors made the effort to address rigorously all comments and questions raised during the first round of reviewing. In particular, my comments and questions were addressed successfuly with changes in figures as well as in the text and I am happy with the clarification/explaination given when necessary.

Altogether, I personally think the manuscript gained in clarity and impact which will for sure satisfy the PLOS Pathogens readership. I'm looking forward to reading this study once published.

**Part II – Major Issues: Key Experiments Required for Acceptance**

Reviewer #1: (No Response)

Reviewer #2: None

Reviewer #3: The authors made some revisions that improved the manuscript, but its length and the amount of data that is presented in the figures, need to be further clarified, since as it is not, it is difficult to follow.

There is confusion regarding the numbering of the tables, as I mentioned in the specific comments.

In general, I think the manuscript is very long and in some cases contains information that is not really adding meaningful information. I think that including 14 figures, 11-12 tables and over 100 references, is too much for a research article in a periodical journal.

Reviewer #4: All the major issues I have raised have now been addressed and relevant explaination were provided when necessary.

**Part III – Minor Issues: Editorial and Data Presentation Modifications**

Reviewer #1: (No Response)

Reviewer #2: 1) For future manuscripts: please move supporting information captions to the end of the manuscript file. From submission guidelines: "List supporting information captions at the end of the manuscript file.". Reviewers 1 and 3 requested this in the previous round of reviews, too.

2) The figure legend for figure 1 should be placed after figure 1 has been quoted in the text (move from line 151 t0 195 in revised version 1). This should be done consistently for all figure legends in the manuscript.

3) Similarly, Table 1 should be moved after the paragraph where it is first referenced in the text. From the journal guidelines: "Insert figure captions in manuscript text, immediately following the paragraph where the figure is first cited (read order)."

4) To improve flow, please move the paragraph from line 275 onwards: "Season had a significant correlation with SARS-CoV-2 positivity with winter having 2.25-fold increased odds (95% CrI 1.47-3.47) compared to fall, and spring having a 2.06-fold increased odds (95% CrI 1.21-3.50) compared to fall. Summer and fall did not show a credible difference for positivity rates from each other." Please move this to where you present figure 2 results (line 243).

Reviewer #3: The minor issues are addressed in the specific comments.

Reviewer #4: All the minor issues I could raised during the first round of reviewing have now be satisfactorily addressed.

PLOS authors have the option to publish the peer review history of their article (what does this mean? ). If published, this will include your full peer review and any attached files.

**Do you want your identity to be public for this peer review?** For information about this choice, including consent withdrawal, please see our Privacy Policy .

Reviewer #1: No

Reviewer #2: No

Reviewer #3: No

Reviewer #4: No

**Figure resubmission:**
---

## [Editor Report · Decision Letter 2]

5 Jan 2025

Dear Dr. Bushman,

We are pleased to inform you that your manuscript 'Evolution of SARS-CoV-2 in White-tailed Deer in Pennsylvania 2021-2024' has been provisionally accepted for publication in PLOS Pathogens.

Best regards,

Shuo Su

Academic Editor

PLOS Pathogens

Ronald Swanstrom

Section Editor

PLOS Pathogens

Sumita Bhaduri-McIntosh

Editor-in-Chief

PLOS Pathogens

orcid.org/0000-0003-2946-9497

Michael Malim

Editor-in-Chief

PLOS Pathogens

orcid.org/0000-0002-7699-2064
---

## [Editor Report · Acceptance letter]

Dear Dr. Bushman,

We are delighted to inform you that your manuscript, "Evolution of SARS-CoV-2 in White-tailed Deer in Pennsylvania 2021-2024," has been formally accepted for publication in PLOS Pathogens.

Best regards,

Sumita Bhaduri-McIntosh

Editor-in-Chief

PLOS Pathogens

orcid.org/0000-0003-2946-9497

Michael Malim

Editor-in-Chief

PLOS Pathogens

orcid.org/0000-0002-7699-2064